# Generative causal explanations
# of black-box classifiers

**Matthew O'Shaughnessy, Gregory Canal, Marissa Connor,**
**Mark Davenport, and Christopher Rozell**
School of Electrical & Computer Engineering
Georgia Institute of Technology

## Abstract

We develop a method for generating causal post-hoc explanations of black-box classifiers based on a learned low-dimensional representation of the data. The explanation is causal in the sense that changing learned latent factors produces a change in the classifier output statistics. To construct these explanations, we design a learning framework that leverages a generative model and information-theoretic measures of causal influence. Our objective function encourages both the generative model to faithfully represent the data distribution and the latent factors to have a large causal influence on the classifier output. Our method learns both global and local explanations, is compatible with any classifier that admits class probabilities and a gradient, and does not require labeled attributes or knowledge of causal structure. Using carefully controlled test cases, we provide intuition that illuminates the function of our objective. We then demonstrate the practical utility of our method on image recognition tasks.[1]

## 1 Introduction

There is a growing consensus among researchers, ethicists, and the public that machine learning models deployed in sensitive applications should be able to *explain* their decisions [1, 2]. A powerful way to make "explain" mathematically precise is to use the language of causality: explanations should identify *causal* relationships between certain data aspects — features which may or may not be semantically meaningful — and the classifier output [3–5]. In this conception, an aspect of the data helps explain the classifier if changing that aspect (while holding other data aspects fixed) produces a corresponding change in the classifier output.

Constructing causal explanations requires reasoning about how changing different aspects of the input data affects the classifier output, but these observed changes are only meaningful if the modified combination of aspects occurs naturally in the dataset. A challenge in constructing causal explanations is therefore the ability to change certain aspects of data samples without leaving the data distribution. In this paper we propose a novel learning-based framework that overcomes this challenge. Our framework has two fundamental components that we argue are necessary to operationalize a causal explanation: a method to *represent and move within the data distribution*, and a *rigorous metric for causal influence* of different data aspects on the classifier output.

To do this, we construct a generative model consisting of a disentangled representation of the data and a generative mapping from this representation to the data space (Figure 1(a)). We seek to learn this disentangled representation in such a way that each factor controls a different aspect of the data, and a subset of the factors have a large causal influence on the classifier output. To formalize this notion of causal influence, we define a structural causal model (SCM) [6] that relates independent

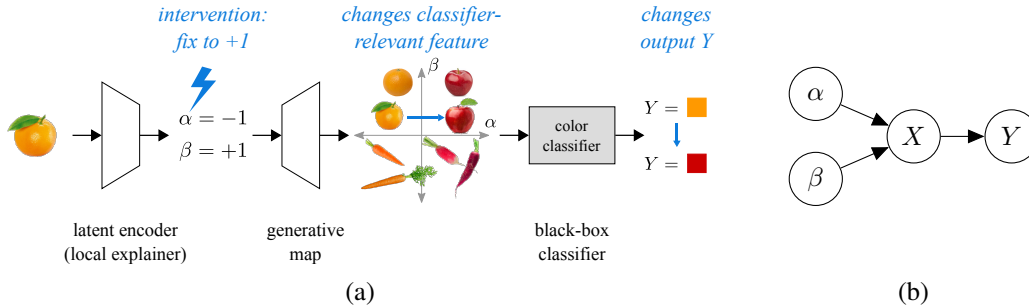

Figure 1: (a) Computational architecture used to learn explanations. Here, the low-dimensional representation $(\alpha, \beta)$ learns to describe the color and shape of inputs. Changing $\alpha$ (color) changes the output of the classifier, which detects the color of the data sample, while changing $\beta$ (shape) does not affect the classifier output. (b) DAG describing our causal model, satisfying principles in Section 3.1.

latent factors defining data aspects, the classifier inputs, and the classifier outputs. Leveraging recent work on information-theoretic measures of causal influence [7, 8], we use the independence of latent factors in the SCM to show that in our framework the causal influence of the latent factors on the classifier output can be quantified simply using mutual information. The crux of our approach is an optimization program for learning a mapping from the latent factors to the data space. The objective ensures that the learned disentangled representation represents the data distribution while simultaneously encouraging a subset of latent factors to have a large causal influence on the classifier output.

A natural benefit of our framework is that the learned disentangled representation provides a rich and flexible vocabulary for explanation. This vocabulary can be more expressive than feature selection or saliency map-based explanation methods: a latent factor, in its simplest form, could describe a single feature or mask of features in input space, but it can also describe much more complex patterns and relationships in the data. Crucially, unlike methods that crudely remove features directly in data space, the generative model enables us to construct explanations that respect the data distribution. This is important because an explanation is only meaningful if it describes combinations of data aspects that naturally occur in the dataset. For example, a loan applicant would not appreciate being told that his loan would have been approved if he had made a negative number of late payments, and a doctor would be displeased to learn that her automated diagnosis system depends on a biologically implausible attribute.

Once the disentangled representation is learned, explanations can be constructed using the generative mapping. Our framework can provide both global and local explanations: a practitioner can understand the aspects of the data that are important to the classifier at large by visualizing the effect in data space of changing each causal factor, and they can determine the aspects that dictated the classifier output for a specific input by observing its corresponding latent values. These visualizations can be much more descriptive than saliency maps, particularly in vision applications.

The major contributions of this work are a new conceptual framework for generating explanations using causal modeling and a generative model (Section 3), analysis of the framework in a simple setting where we can obtain analytical and intuitive understanding (Section 4), and a brief evaluation of our method applied to explaining image recognition models (Section 5).

## 2 Related work

We focus on methods that generate *post-hoc* explanations of black-box classifiers. While post-hoc explanations are typically categorized as either global (explaining the entire classifier mechanism) or local (explaining the classification of a particular datapoint) [9], our framework joins a smaller group of methods that globally learn a model that can be then used to generate local explanations [10–13].

**Forms of explanation.** Post-hoc explanations come in varying forms. Some methods learn an interpretable model such as a decision tree that *approximates the black-box* either globally [14–16] or locally [17–20]. A larger class of methods create local explanations directly in the data space,

performing *feature selection* or creating *saliency maps* using classifier gradients [21–25] or by training a new model [10]. A third category of methods generate *counterfactual data points* that describe how inputs would need to be altered to produce a different classifier output [26–32]. Other techniques identify the *points in the training set* most responsible for a particular classifier output [33, 34]. Our framework belongs to a separate class of methods whose explanations consist of a low-dimensional set of *latent factors* that describe different aspects (or "concepts") of the data. These latent factors form a rich and flexible vocabulary for both global and local explanations, and provide a means to represent the data distribution. Unlike some methods that learn concepts using labeled attributes [35, 36], we do not require side information defining data aspects; rather, we visualize the learned aspects using a generative mapping to the data space as in [37–39]. This type of latent factor explanation has also been used in the construction of self-explaining neural networks [37, 40].

**Causality in explanation.** Because explanation methods seek to answer "why" and "how" questions that use the language of cause and effect [3, 4], causal reasoning has played an increasingly important role in designing explanation frameworks [5]. (For similar reasons, causality has played a prominent part in designing metrics for fairness in machine learning [41–45].) Prior work has quantified the impact of features in data space by using Granger causality [13], a priori known causal structure [46, 36], an average or individual causal effect metric [47, 19], or by applying random valued-interventions [48]. Other work generates causal explanations by performing interventions in different network layers [49], using latent factors built into a modified network architecture [38], or using labeled examples of human-interpretable latent factors [50].

Generative models have been used to compute interventions that respect the data distribution [51, 36, 19, 52], a key idea in this paper. Our work, however, is most similar to methods using generative models whose explanations use notions of causality and are constructed directly from latent factors. Goyal et al. compute the average causal effect (ACE) of human-interpretable concepts on the classifier [50], but require labeled examples of the concepts and suffer from limitations of the ACE metric [8]. Harradon et al. construct explanations based on latent factors, but these explanations are specific to neural network classifiers and require knowledge of the classifier network architecture [38]. Our method is unique in constructing a framework from principles of causality that generates latent factor-based explanations of black-box classifiers without requiring side information.

**Disentanglement perspective.** Our method can also be interpreted as a *disentanglement* procedure [53, 54] supervised by classifier output probabilities. Unlike work that encourages a one-to-one correspondence between individual latent factors and semantically meaningful features (i.e., "data generating factors"), we aim to separate the latent factors that are relevant to the classifier's decision from those that are irrelevant. We outline connections to this literature in more detail in Section 3.5.

## 3 Methods

Our goal is to explain a black-box classifier $f\colon \mathcal{X} \to \mathcal{Y}$ that takes data samples $X \in \mathcal{X}$ and assigns a probability to each class $Y \in \{1, \ldots, M\}$ (i.e., $\mathcal{Y}$ is the $M$-dimensional probability simplex). We assume that the classifier also provides the gradient of each class probability with respect to the classifier input.

Our explanations take the form of a low-dimensional and independent set of "causal factors" $\alpha \in \mathbb{R}^K$ that, when changed, produce a corresponding change in the classifier output statistics. We also allow for additional independent latent factors $\beta \in \mathbb{R}^L$ that contribute to representing the data distribution but need not have a causal influence on the classifier output. Together, $(\alpha, \beta)$ constitute a low-dimensional representation of the data distribution $p(X)$ through the generative mapping $g\colon \mathbb{R}^{K+L} \to \mathcal{X}$. The generative mapping is learned so that the explanatory factors $\alpha$ have a large causal influence on $Y$, while $\alpha$ and $\beta$ together faithfully represent the data distribution (i.e., $p(g(\alpha, \beta)) \approx p(X)$). The $\alpha$ learned in this manner can be interpreted as aspects *causing* $f$ to make classification decisions [6].

To learn a generative mapping with these characteristics, we need to define (i) a model of the causal relationship between $\alpha$, $\beta$, $X$, and $Y$, (ii) a metric to quantify the causal influence of $\alpha$ on $Y$, and (iii) a learning framework that maximizes this influence while ensuring that $p(g(\alpha, \beta)) \approx p(X)$.

### 3.1 Causal model

We first define a directed acyclic graph (DAG) describing the relationship between $(\alpha, \beta)$, $X$, and $Y$, which will allow us to derive a metric of causal influence of $\alpha$ on $Y$. We propose the following principles for selecting this DAG:

(1) **The DAG should describe the functional (causal) structure of the data, not simply the statistical (correlative) structure.** This principle allows us to interpret the DAG as a structural causal model (SCM) [6] and interpret our explanations causally.

(2) **The explanation should be derived from the classifier output $Y$, not the ground truth classes.** This principle affirms that we seek to understand the action of the classifier, not the ground truth classes.

(3) **The DAG should contain a (potentially indirect) causal link from $X$ to $Y$.** This principle ensures that our causal model adheres to the functional operation of $f \colon X \to Y$.

Based on these principles, we adopt the DAG shown in Figure 1(b). Note that the difference in the roles played by $\alpha$ and $\beta$ is subtle and not apparent from the DAG alone: the difference arises from the fact that the functional relationship defining the causal connection $X \to Y$ is $f$, which by construction uses only features of $X$ that are controlled by $\alpha$. In other words, interventions on both $\alpha$ and $\beta$ produce changes in $X$, but only interventions on $\alpha$ produce changes in $Y$. A key feature of this DAG is that the latent factors $(\alpha, \beta)$ are independent, which we enforce with an isotropic prior when learning the generative mapping. This independence improves the parsimony and interpretability of the learned disentangled representation (see Appendix A). It also results in our metric for causal influence simplifying to mutual information. Importantly, unlike methods that *assume* independence of features in data space (e.g., [48, 17, 23, 25]), our framework *intentionally learns* independent latent factors.

### 3.2 Metric for causal influence

We now derive a metric $\mathcal{C}(\alpha, Y)$ for the causal influence of $\alpha$ on $Y$ using the DAG in Figure 1(b). A satisfactory measure of causal influence in our application should satisfy the following principles:

(1) **The metric should completely capture functional dependencies.** This principle allows us to capture the complete causal influence of $\alpha$ on $Y$ through the generative mapping $g$ and classifier $f$, which may both be defined by complex and nonlinear functions such as neural networks.

(2) **The metric should quantify indirect causal relationships between variables.** This principle allows us to quantify the indirect causal relationship between $\alpha$ and $Y$.

Principle 1 eliminates common metrics such as the average causal effect (ACE) [55] and analysis of variance (ANOVA) [56], which capture only causal relationships between first- and second-order statistics, respectively [8]. Recent work has overcome these limitations by using information-theoretic measures [7, 8, 57]. Of these, we select the *information flow* measure of [7] to satisfy Principle 2 because it is node-based, naturally accommodating our goal of quantifying the causal influence of $\alpha$ on $Y$.

The information flow metric adapts the concept of mutual information typically used to quantify *statistical* influence to quantify *causal* influence by the observational distributions in the standard definition of conditional mutual information with interventional distributions:

**Definition 1** (Ay and Polani 2008 [7]). *Let $U$ and $V$ be disjoint subsets of nodes. The* information flow *from $U$ to $V$ is*

$$I(U \to V) := \int_U p(u) \int_V p(v \mid do(u)) \log \frac{p(v \mid do(u))}{\int_{u'} p(u')p(v \mid do(u'))du'} dV\, dU, \qquad (1)$$

*where $do(u)$ represents an intervention in a causal model that fixes $u$ to a specified value regardless of the values of its parents [6].*

The independence of $(\alpha, \beta)$ makes it simple to show that information flow and mutual information coincide in our DAG:

**Proposition 2** (Information flow in our DAG). *The information flow from $\alpha$ to $Y$ in the DAG of Figure 1(b) coincides with the mutual information between $\alpha$ and $Y$. That is, $I(\alpha \to Y) = I(\alpha; Y)$, where mutual information is defined as $I(\alpha; Y) = \mathbb{E}_{\alpha, Y}\left[\log \frac{p(\alpha, Y)}{p(\alpha)p(Y)}\right]$.*

---

**Algorithm 1** Principled procedure for selecting $(K, L, \lambda)$.

---

1: Initialize $K, L, \lambda = 0$. Optimizing only $\mathcal{D}$, increase $L$ until objective plateaus.
2: **repeat** increment $K$ and decrement $L$. Increase $\lambda$ until $\mathcal{D}$ approaches value from Step 1.
3: **until** $\mathcal{C}$ reaches plateau. Use $(K, L, \lambda)$ from immediately before plateau was reached.

---

The proof, which follows easily from the rules of do-calculus [6, Thm. 3.4.1], is provided in Appendix C.1. Based on this result, we use

$$\mathcal{C}(\alpha, Y) = I(\alpha; Y) \tag{2}$$

to quantify the causal influence of $\alpha$ on $Y$. This metric, derived in our work from principles of causality using the DAG in Figure 1(b), has also been used to select informative features in other work on explanation [58, 11, 40, 59–61]. Our framework, then, generates explanations that benefit from both causal and information-theoretic perspectives. Note, however, that the validity of the causal interpretation is predicated on our modeling decisions; mutual information is in general a correlational, not causal, metric.

Other variants of (conditional) mutual information are also compatible with our development. These variants retain causal interpretations, but produce explanations of a slightly different character. For example, $\sum_{i=1}^{K} I(\alpha_i; Y)$ and $I(\alpha; Y \mid \beta)$ (the latter corresponding to the information flow of $\alpha$ on $Y$ when "imposing" $\beta$ in [7]) encourage interactions between the explanatory features to generate $X$. These variants are described and analyzed in more detail in Appendices A and B.

### 3.3 Optimization framework

We now turn to our goal of learning a generative mapping $g: (\alpha, \beta) \to X$ such that $p(g(\alpha, \beta)) \approx p(X)$, the $(\alpha, \beta)$ are independent, and $\alpha$ has a large causal influence on $Y$. We do so by solving

$$\underset{g \in G}{\arg\max} \quad \mathcal{C}(\alpha, Y) + \lambda \cdot \mathcal{D}\left(p(g(\alpha, \beta)), p(X)\right), \tag{3}$$

where $g$ is a function in some class $G$, $\mathcal{C}(\alpha, Y)$ is our metric for the causal influence of $\alpha$ on $Y$ from (2), and $\mathcal{D}(p(g(\alpha, \beta)), p(X))$ is a measure of the similarity between $p(g(\alpha, \beta))$ and $p(X)$.

The use of $\mathcal{D}$ is a crucial feature of our framework because it forces $g$ to produce samples that are in the data distribution $p(X)$. Without this property, the learned causal factors could specify combinations of aspects that do not occur in the dataset, providing little value for explanation. The specific form of $\mathcal{D}$ is dependent on the class of decoder models $G$. In this paper we focus on two specific instantiations of $G$. Section 4 takes $G$ to be the set of linear mappings with Gaussian additive noise, using negative KL divergence for $\mathcal{D}$. This setting allows us to provide more rigorous intuition for our model. Section 5 adopts the variational autoencoder (VAE) framework shown in Figure 1(a), parameterizing $G$ by a neural network and using a variational lower bound [62] as $\mathcal{D}$.

### 3.4 Training procedure

In practice, we maximize the objective (3) using Adam [63], computing a sample-based estimate of $\mathcal{C}$ at each iteration. The sampling procedure is detailed in Appendix D. Training our causal explanatory model requires selecting $K$ and $L$, which define the number of latent factors, and $\lambda$, which trades between causal influence and data fidelity in our objective. A proper selection of these parameters should set $\lambda$ sufficiently large so that the distributions $p(X \mid \alpha, \beta)$ used to visualize explanations lie in the data distribution $p(X)$, but not so high that the causal influence term is overwhelmed.

To properly navigate this trade-off it is instructive to view (3) as a constrained problem [64] in which $\mathcal{C}$ is maximized subject to an upper bound on $\mathcal{D}$. Algorithm 1 provides a principled method for parameter selection based on this idea. Step 1 selects the total number of latent factors needed to adequately represent $p(X)$ using only noncausal factors. Steps 2-3 then incrementally convert noncausal factors into causal factors until the total explanatory value of the causal factors (quantified by $\mathcal{C}$) plateaus. Because changing $K$ and $L$ affects the relative weights of the causal influence and data fidelity terms, $\lambda$ should be increased after each increment to ensure that the learned representation continues to satisfy the data fidelity constraint.

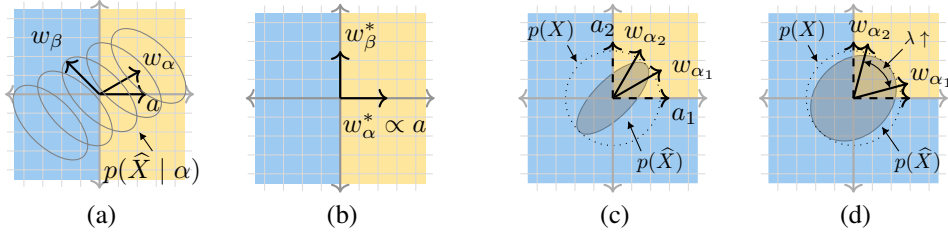

Figure 2: Explaining simple classifiers in $\mathbb{R}^2$. (a) Visualizing the conditional distribution $p(\widehat{X} \mid \alpha)$ provides intuition for the linear-Gaussian model. (b) Linear classifier with yellow encoding high probability of $y = 1$ (right side), and blue encoding high probability of $y = 0$ (left side). Proposition 3 shows that the optimal solution to (3) is $w_\alpha^* \propto a$ and $w_\beta^* \perp w_\alpha^*$ for $\lambda > 0$. (c-d) For the "and" classifier, varying $\lambda$ trades between causal alignment and data representation.

## 3.5 Disentanglement perspective

Disentanglement procedures seek to learn low-dimensional data representations in which latent factors correspond to data aspects that concisely and independently describe high dimensional data [53, 54]. Although some techniques perform unsupervised disentanglement [65–67], it is common to use side information as a supervisory signal.

Because our goal is explanation, our main objective is to separate classifier-relevant and classifier-irrelevant aspects. Our framework can be thought of as a disentanglement procedure with two distinguishing features:

First, we use classifier probabilities to aid disentanglement. This is similar in spirit to disentanglement methods that incorporate grouping or class labels as side information by modifying the VAE training procedure [68], probability model [69], or loss function [70]. Although these methods could be adapted for explanation using classifier-based groupings, our method intelligently uses classifier *probabilities* and gradients.

Second, we develop our framework from a causal perspective. Suter et al. also develop a disentanglement procedure from principles of causality [71], casting the disentanglement task as learning latent factors that correspond to parent-less causes in the generative structural causal model. Unlike this framework, we assume that the latent factors are independent based on properties of the VAE evidence lower bound. We then use this fact to show that the commonly-used MI metric measures *causal* influence of $\alpha$ on $Y$ using the information flow metric of [7].

This provides a causal interpretation for information-based disentanglement methods such as In-foGAN [66] (which adds a term similar to $I(\alpha; X)$ to the VAE objective). Encouragement of independence in latent factors plays an important role in much work on disentanglement (e.g., [65, 66, 72]); priors that better encourage independence could be applied in our framework to increase the validity of our proposed causal graph.

## 4  Analysis with linear-Gaussian generative map

We first consider the instructive setting in which a linear generative mapping is used to explain simple classifiers with decision boundaries defined by hyperplanes. This setting admits geometric intuition and basic analysis that illuminates the function of our objective.

In this section we define the data distribution as isotropic normal in $\mathbb{R}^N$, $X \sim \mathcal{N}(0, I)$ (but note that elsewhere in the paper we make no assumptions on the data distribution). Let $(\alpha, \beta) \sim \mathcal{N}(0, I)$, and consider the following generative model to be used for constructing explanations:

$$g(\alpha, \beta) = [W_\alpha \quad W_\beta] \begin{bmatrix} \alpha \\ \beta \end{bmatrix} + \varepsilon,$$

where $W_\alpha \in \mathbb{R}^{N \times K}$, $W_\beta \in \mathbb{R}^{N \times L}$, and $\varepsilon \sim \mathcal{N}(0, \gamma I)$. We illustrate the behavior of our method applied with this generative model on two simple binary classifiers ($Y \in \{0, 1\}$).

**Linear classifier.** Consider first a linear separator $p(y = 1 \mid x) = \sigma(a^T x)$, where $a \in \mathbb{R}^N$ denotes the decision boundary normal and $\sigma$ is a sigmoid function (visualized in $\mathbb{R}^2$ in Figure 2(a)). With a single causal and single noncausal factor ($K = L = 1$), learning an explanation consists of finding the $w_\alpha, w_\beta \in \mathbb{R}^2$ that maximize (3). Intuitively, we expect $w_\alpha$ to align with $a$ because this direction allows $\alpha$ to produce the largest change in classifier output statistics. This can be seen by considering the distribution $p(\widehat{X} \mid \alpha)$ depicted in Figure 2(a), where we denote $\widehat{X} = g(\alpha, \beta)$ for convenience. Since the generative model is linear-Gaussian, varying $\alpha$ translates $p(\widehat{X} \mid \alpha)$ along the direction $w_\alpha$. When this direction is more aligned with the classifier normal $a$, interventions on $\alpha$ cause a larger change in classifier output by moving $p(\widehat{X} \mid \alpha)$ across the decision boundary. Because the data distribution is isotropic, we expect $\mathcal{D}$ to achieve its maximum when $w_\beta$ is orthogonal to $w_\alpha$, allowing $w_\alpha$ and $w_\beta$ to perfectly represent the data distribution. By combining these two insights, we see that the solution of (3) is given by $w_\alpha^* \propto a$ and $w_\beta^* \perp w_\alpha^*$ (Figure 2(b)).

This intuition is formalized in the following proposition, where for analytical convenience we use the (sigmoidal) normal cumulative distribution function as the classifier nonlinearity $\sigma$:

**Proposition 3.** *Let* $\mathcal{X} = \mathbb{R}^N$, $K = 1$, $L = N - 1$, *and* $p(Y = 1 \mid x) = \sigma(a^T x)$, *where* $\sigma$ *is the normal cumulative distribution function. Suppose that the columns of* $W = [w_\alpha \; W_\beta]$ *are normalized to magnitude* $\sqrt{1 - \gamma}$ *with* $\gamma < 1$. *Then for any* $\lambda > 0$ *and for* $\mathcal{D}(p(\widehat{X}), p(X)) = -\mathrm{D_{KL}}(p(X) \parallel p(\widehat{X}))$, *the objective* (3) *is maximized when* $w_\alpha \propto a$, $W_\beta^T a = 0$, *and* $W_\beta^T W_\beta = (1 - \gamma)I$.

The proof, which is listed in Appendix C.2, follows geometric intuition for the behavior of $\mathcal{C}$. This result verifies our objective's ability to construct explanations with our desired properties: the causal factor learns the direction in which the classifier output changes, and the complete set of latent factors represent the data distribution.

**"And" classifier.** Now consider the slightly more complex "and" classifier parameterized by two orthogonal hyperplane normals $a_1, a_2 \in \mathbb{R}^2$ (Figure 2(c)) given by $p(Y = 1 \mid x) = \sigma(a_1^T x) \cdot \sigma(a_2^T x)$. This classifier assigns a high probability to $Y = 1$ when both $a_1^T x > 0$ and $a_2^T x > 0$. Here we use $K = 2$ causal factors and $L = 0$ noncausal factors to illustrate the role of $\lambda$ in trading between the terms in our objective. In this setting, learning an explanation entails finding the $w_{\alpha_1}, w_{\alpha_2} \in \mathbb{R}^2$ that maximize (3).

Figure 2(c-d) depicts the effect of $\lambda$ on the learned $w_{\alpha_1}, w_{\alpha_2}$ (see Appendix B for empirical visualizations). Unlike in the linear classifier case, when explaining the "and" classifier there is a tradeoff between the two terms in our objective: the causal influence term encourages both $w_{\alpha_1}$ and $w_{\alpha_2}$ to point towards the upper right-hand quadrant of the data space, the direction that produces the largest variation in class output probability. On the other hand, the isotropy of the data distribution results in the data fidelity term encouraging orthogonality between the factor directions. Therefore, when $\lambda$ is small the causal effect term dominates, aligning the causal factors to the upper right-hand quadrant of the data space (Figure 2(c)). As $\lambda$ increases (Figure 2(d)), the larger weight on the data fidelity term encourages orthogonality between the factor directions so that $p(\widehat{X})$ more closely approximates $p(X)$. This example illustrates how $\lambda$ must be selected carefully to represent the data distribution while learning meaningful explanatory directions (see Section 3.4).

## 5   Experiments with VAE architecture

In this section we generate explanations of CNN classifiers trained on image recognition tasks, letting $G$ be a set of neural networks and adopting the VAE architecture shown in Figure 1(a) to learn $g$.

**Qualitative results.** We train a CNN classifier with two convolutional layers followed by two fully connected layers on MNIST 3 and 8 digits, a common test setting for explanation methods [25, 13]. Using the parameter tuning procedure described in Algorithm 1, we select $K = 1$ causal factor, $L = 7$ noncausal factors, and $\lambda = 0.05$. Figure 3(a) shows the global explanation for this classifier and dataset, which visualizes how $g(\alpha, \beta)$ changes as $\alpha$ is modified. We observe that $\alpha$ controls the features that differentiate the digits 3 and 8, so changing $\alpha$ changes the classifier output while preserving stylistic features irrelevant to the classifier such as skew and thickness. By contrast, Figures 3(b-d) show that changing each $\beta_i$ affects stylistic aspects such as thickness and skew but not

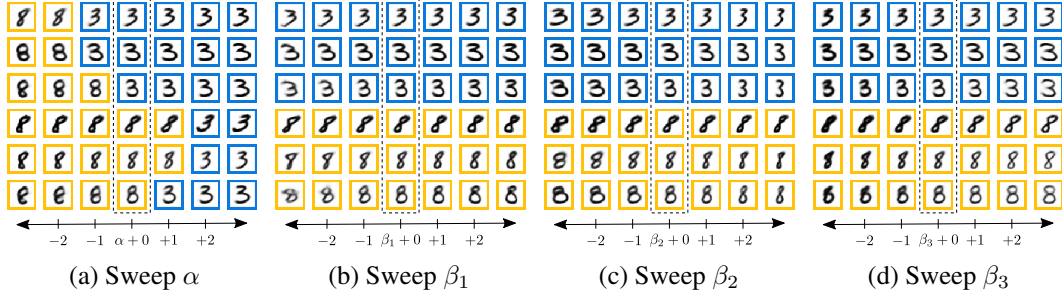

|  (a) Sweep $\alpha$ | (b) Sweep $\beta_1$ | (c) Sweep $\beta_2$ | (d) Sweep $\beta_3$ |

Figure 3: Visualizations of learned latent factors. (a) Changing the causal factor $\alpha$ provides the global explanation of the classifier. Images in the center column of each grid are reconstructed samples from the validation set; moving left or right in each row shows $g(\alpha, \beta)$ as a single latent factor is varied. Changing the learned causal factor $\alpha$ affects the classifier output (shown as colored outlines). (b-d) Changing the noncausal factors $\{\beta_i\}$ affects stylistic aspects such as thickness and skew but does not affect the classifier output.

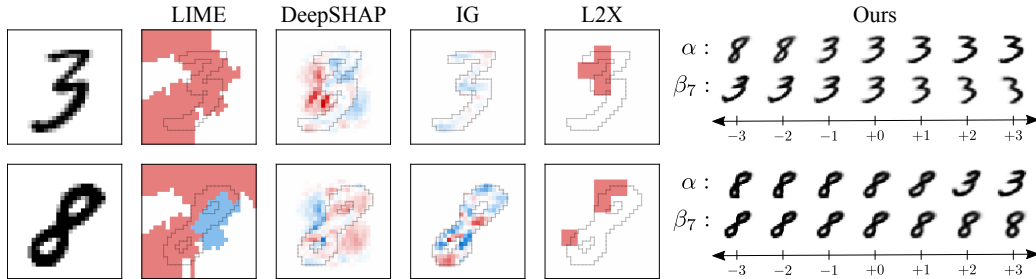

Figure 4: Compared to popular explanation techniques that generate saliency map-based explanations, our explanations consist of learned aspect(s) of the data, visualized by sweeping the associated latent factors (remaining latent factor sweeps are shown in Appendix E.2). Our explanations are able to differentiate causal aspects (pixels that define 3 from 8) from purely stylistic aspects (here, rotation).

the classifier output. Details of the experimental setup and training procedure are listed in Appendix E.1 along with additional results.

**Comparison to other methods.** Figure 4 shows the explanations generated by several popular competitors: LIME [17], DeepSHAP [25], Integrated Gradients (IG) [24], and L2X [11]. Each of these methods generates explanations that quantify a notion of relevance of (super)pixels to the classifier output, visualizing the result with a saliency map. While this form of explanation can be appealing for its simplicity, it fails to capture more complex relationships between pixels. For example, saliency map explanations cannot differentiate the "loops" that separate the digits 3 and 8 from other stylistic factors such as thickness and rotation present in the same (super)pixels. Our explanations overcome this limitation by instead visualizing latent factors that control different aspects of the data. This is demonstrated on the right of Figure 4, where latent factor sweeps show the difference between classifier-relevant and purely stylistic aspects of the data. Observe that $\alpha$ controls data aspects used by the classifier to differentiate between classes, while the noncausal factor controls rotation. Appendix E.2 visualizes the remaining noncausal factors and details the experimental setup.

**Quantitative results.** We next learn explanations of a CNN trained to classify t-shirt, dress, and coat images from the Fashion MNIST dataset [73]. Following the parameter selection procedure of Algorithm 1, we select $K = 2$, $L = 4$, and $\lambda = 0.05$. We evaluate the efficacy of our explanations in this setting using two quantitative metrics. First, we compute the information flow (1) from each latent factor to the classifier output $Y$. Figure 5(a) shows that, as desired, the information flow from $\alpha$ to $Y$ is large while the information flow from $\beta$ to $Y$ is small. Second, we evaluate the reduction in classifier accuracy after individual aspects of the data are removed by fixing a single latent factor in each validation data sample to a different random value drawn from the prior $\mathcal{N}(0, 1)$. This test is frequently used as a metric for explanation quality; our method has the advantage of allowing us to remove certain data aspects while remaining in-distribution rather than crudely removing features

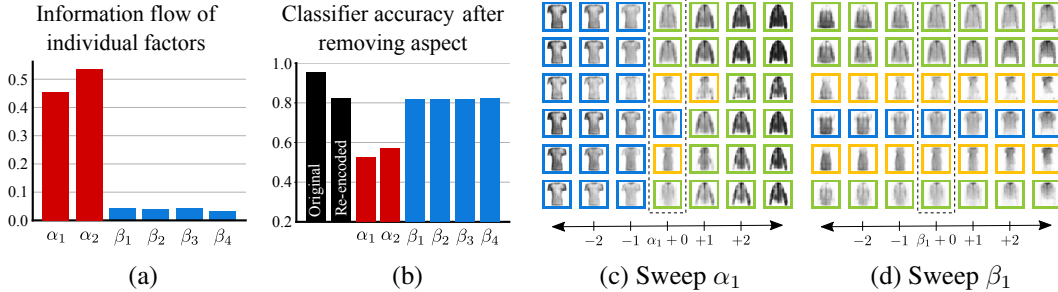

Figure 5: (a) Information flow (1) of each latent factor on the classifier output statistics. (b) Classifier accuracy when data aspects controlled by individual latent factors are removed (original: accuracy on validation set; re-encoded: classifier accuracy on validation set encoded and reconstructed by VAE), showing that learned causal factors (but not noncausal factors) control data aspects relevant to the classifier. (c-d) Modifying $\alpha_1$ changes the classifier output, while modifying $\beta_1$ does not.

by masking (super)pixels [74]. Figure 5(b) shows this reduction in classifier accuracy. Observe that changing aspects controlled by learned causal factors indeed significantly degrades the classifier accuracy, while removing aspects controlled by noncausal factors has only a negligible impact on the classifier accuracy. Figure 5(c-d) visualizes the aspects learned by $\alpha_1$ and $\beta_1$. As before, only the aspects of the data controlled by $\alpha$ are relevant to the classifier: changing $\alpha_1$ produces a change in the classifier output, while changing $\beta_1$ affects only aspects that do not modify the classifier output. Appendix E.3 contains details on the experimental setup and complete results.

## 6 Discussion

The central contribution of our paper is a generative framework for learning a rich and flexible vocabulary to explain a black-box classifier, and a method that uses this vocabulary and causal modeling to construct explanations. Our derivation from a causal model allows us to learn explanatory factors that have a causal, not correlational, relationship with the classifier, and the information-theoretic measure of causality that we adapt allows us to completely capture complex causal relationships. Our use of a generative framework to learn independent latent factors that describe different aspects of the data allows us to ensure that our explanations respect the data distribution.

Applying this framework to practical explanation tasks requires selecting a generative model architecture, and then training this generative model using data relevant to the classification task. The data used to train the explainer may be the original training set of the classifier, but more generally it can be any dataset; the resulting explanation will reveal the aspects in that specific dataset that are relevant to the classifier. The user must also select a generative model $g$ with appropriate capacity. Underestimating this capacity could reduce the effectiveness of the resulting explanations, while overestimating this capacity will needlessly increase the training cost. We explore this selection further in Appendix F both empirically and by using results from [75] to show how the value of $I(\alpha; Y)$ can be interpreted as a "certificate" of sufficient generative model capacity.

Our framework combining generative and causal modeling is quite general. Although we focused on the use of learned data aspects to generate explanations by visualizing the effect of modifying learned causal factors, the learned representation could also be used to generate counterfactual explanations — minimal perturbations of a data sample that change the classifier output [29, 3]. Our framework would address two common challenges in counterfactual explanation: because we can optimize over a low-dimensional set of latent factors, we avoid a computationally infeasible search in input space, and because each point in space maps to an in-distribution data sample, our model naturally ensures that perturbations result in a valid data point. Another promising avenue for future work is relaxing the independence structure of learned causal factors. Although this would result in a more complex expression for information flow, the sampling procedure we use to compute causal effect would generalize naturally; the more challenging obstacle would be learning latent factors with nontrivial causal structure. Finally, techniques that make the classifier-relevant latent factors more interpretable or better communicate the aspects controlled by each latent factor to humans would improve the quality of our generated explanations.

## Broader impacts

Explanation methods have the potential to play a major role in enabling the safe and fair deployment of machine learning systems [2, 76], and explainability is a oft-mentioned constraint in their legal and ethical analysis. Policy discussions about machine learning have increasingly turned to principles of transparency and fairness [77], with some legal scholars arguing that the 2016 European General Data Protection Regulation (GDPR) contains a "right to explanation" [78], and recent G20 and OECD recommendations both identifying "transparency and explainability" as important principles for the development of machine learning algorithms [79, 80].

The growing literature on explainability that our work contributes to has the potential to improve the transparency and fairness of machine learning systems and increase the level of trust users place in their decisions. Yet these explanation methods, often built from complex and nontransparent components and each proposing subtly different notions of explanation, also risk providing deceptively incomplete understanding of systems used in sensitive applications, or providing false assurances of fairness and lack of bias (see, e.g., [81]). This criticism may be especially true for our method, which constructs explanations using neural networks that are themselves difficult to understand. For the explanation literature to have a positive impact, it is necessary for explanations to be easily yet precisely understood by the nontechnical generalists deploying and regulating machine learning systems. We believe that causal perspective used in this work is valuable in this regard because causality has been identified as a vocabulary appropriate for translating technical concepts to psychological [3] and legal frameworks [2, 29]. We also believe our analysis with simple models is important because it endows our explanations with some theoretical grounding. However, a critical need remains for more interdisciplinary research examining how end users understand the outputs of explanation tools (e.g., [82]) and how technical tools can be brought to bear to address identified deficiencies.

## Acknowledgments and Disclosure of Funding

This work was supported by NSF grant CCF-1350954, a gift from the Alfred P. Sloan Foundation, and the National Defense Science & Engineering Graduate (NDSEG) Fellowship.

## Footnotes

[1]Code is available at `https://github.com/siplab-gt/generative-causal-explanations`.

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
