[Supplementary Material]

# A    Intuition for and variants of causal influence metric

**Intuition for causal influence objective.** To better understand the causal portion of our objective (4), we use standard identities to decompose it as

$$\mathcal{C} = I(Y; \alpha) = H(Y) - \mathbb{E}_\alpha[H(Y \mid \alpha)], \qquad (4)$$

where

$$p(y \mid \alpha) = \int_\beta \int_x p(y \mid x) p(x \mid \alpha, \beta) p(\beta) dx d\beta. \qquad (5)$$

The conditional distribution (5) can be interpreted as the probability of $Y = y$ for a fixed value of $\alpha$, averaged over the values of $\beta$. The decomposition in (4) therefore shows that $\mathcal{C}$ is the *reduction in uncertainty about $Y$ provided by knowledge of $\alpha$*, where this reduction is measured in a global sense in that the effect of $\beta$ is averaged together to produce a single probability estimate for $Y$ and fixed $\alpha$.

As an example, consider the color classifier and generative mapping shown in Figure 1(a), in which $f$ classifies based on color. The first term in (4) represents how similar the classifier output is for all objects in the training set. The second term represents how similar the classifier output for groups of objects is, on average, after being grouped by $\alpha$. A large $\mathcal{C} = I(\alpha; Y)$ means that grouping by $\alpha$ significantly increases the confidence the classifier has that objects in each group are of the same class. In this case, grouping by $\alpha = $ 'color' has a much larger effect on the classifier output — and therefore results in a larger $\mathcal{C}$ — than grouping by $\alpha = $ 'shape' would, since grouping the objects by color results in the classifier gaining much more confidence that each group shares the same class.

**Variants of causal objective.** Consider the following variants of the *joint, unconditional* objective $\mathcal{C} = I(\alpha; Y)$, our measure of causal influence from Section 3.2:

1. *Independent, unconditional:* $\mathcal{C}_{iu} = \frac{1}{K} \sum_i I(\alpha_i; Y)$
2. *Independent, conditional:* $\mathcal{C}_{ic} = \frac{1}{K} \sum_i I(\alpha_i; Y \mid \alpha_{\neg i}, \beta)$, where $\alpha_{\neg i} = \{\alpha_j\}_{j \neq i}$
3. *Joint, conditional:* $\mathcal{C}_{jc} = I(\alpha; Y \mid \beta)$

Each objective variant gives rise to a classifier explanation that has a causal interpretation, but as we will show, the *character* of each is subtly different. The following proposition begins to explore these differences by relating them using information-theoretic quantities.

**Proposition 4** (Relationship between candidate causal objectives). *The following hold in the DAG of Figure 1(b):*

(a) $\mathcal{C} = \mathcal{C}_{iu} + \frac{1}{K} \sum_{i=1}^K I(\alpha_{\neg i}; Y \mid \alpha_i)$.

(b) $\mathcal{C}_{jc} = \mathcal{C}_{ic} + \frac{1}{K} \sum_{i=1}^K I(\alpha_{\neg i}; Y \mid \beta)$.

(c) $\mathcal{C}_{jc} = \mathcal{C} + I(\alpha; \beta \mid Y)$.

(d) $\mathcal{C}_{ic} = \mathcal{C}_{iu} + \frac{1}{K} \sum_i I(\alpha_i; \alpha_{\neg i}, \beta \mid Y)$.

These relationships are depicted visually in Figure 6 and proved in Appendix C.3. Note that only (c) and (d) use the independence of the latent variables in our DAG. The "adjustment factors" that relate the objective variants can be interpreted as follows:

1. By conditioning on other latent factors (i.e., using $\mathcal{C}_{ic}$, $\mathcal{C}_{iu}$, or $\mathcal{C}_{jc}$ rather than $\mathcal{C}$) we include the "adjustment factor" $\frac{1}{K} \sum_i I(\alpha_i; \alpha_{\neg i}, \beta \mid Y)$ (in the "independent" case) or $I(\alpha; \beta \mid Y)$ (in the "joint" case) in the objective. These terms encourage complex interactions between latent factors within each group of similarly-classified points. On the one hand, the stastistical pattern that these terms encourage arises naturally from the DAG in Figure 1(b): although the latent factors are independent, conditioning on $Y$ renders them dependent. This conditional dependence pattern is often referred to as Berkson's paradox or the "explaining away" phenomenon. To illustrate this concept, consider a classifier that classifies paintings at an auction as $Y \in \{$'sold', 'not sold'$\}$ based on the learned latent factors $z_1 = $ 'beautiful' and $z_2 = $ 'historical value', which we assume to be independent. Once $Y$ is known, however, $z_1$ and $z_2$ are rendered dependent: learning that a sold painting does not have historical

| independent, unconditional $\mathcal{C}_{iu} = \frac{1}{K}\sum_i I(\alpha_i; Y)$ | $+\frac{1}{K}\sum_i I(\alpha_{\neg i}; Y \mid \alpha_i)$ → | joint, unconditional $\mathcal{C} = I(\alpha; Y)$ |

$+\frac{1}{K}\sum_i I(\alpha_i; \alpha_{\neg i}, \beta \mid Y)$ ↓    $+I(\alpha; \beta \mid Y)$ ↓

| independent, conditional $\mathcal{C}_{ic} = \frac{1}{K}\sum_i I(\alpha_i; Y \mid \alpha_{\neg i}, \beta)$ | $+\frac{1}{K}\sum_i I(\alpha_{\neg i}; Y \mid \beta)$ → | joint, conditional $\mathcal{C}_{jc} = I(\alpha; Y \mid \beta)$ |

Figure 6: Graphical representation of relationships between causal objective variants derived from Proposition 4.

value would allow us to infer that it is likely to be beautiful. On the other hand, we do not in general expect that our learned latent factors, which we encourage to be independent, will correspond to semantically meaningful features, so we may not expect them to fit this "explaining away" conditional dependence pattern.

2. By jointly considering the causal factors $\alpha$ rather than summing the causal influence of each $\alpha_i$ (i.e., by using $\mathcal{C}$ rather than $\mathcal{C}_{iu}$, or $\mathcal{C}_{jc}$ rather than $\mathcal{C}_{ic}$) we include the "adjustment factor" $\frac{1}{K}\sum_{i=1}^{K} I(\alpha_{\neg i}; Y \mid \alpha_i)$ in the objective. This term encourages each learned causal factor to make the remaining causal factors more predictable given the classifier output $Y$, encouraging *interactions* between latent factors to have an effect on the classifier output probability. We consider this to be positive, but using an independent objective might aid in visualizing the relationship between the latent space and data space.

The next section provides more intuition for these objectives in the context of the linear-Gaussian generative map and simple classifiers introduced in Section 4.

(a) $\theta(w_\alpha) = 0°$, $\theta(w_\beta) = 0°$    (b) $\theta(w_\alpha) = 45°$, $\theta(w_\beta) = 0°$    (c) $\theta(w_\alpha) = 90°$, $\theta(w_\beta) = 0°$

(d) $\theta(w_\alpha) = 0°$, $\theta(w_\beta) = 45°$    (e) $\theta(w_\alpha) = 45°$, $\theta(w_\beta) = 45°$    (f) $\theta(w_\alpha) = 90°$, $\theta(w_\beta) = 45°$

(g) $\theta(w_\alpha) = 0°$, $\theta(w_\beta) = 90°$    (h) $\theta(w_\alpha) = 45°$, $\theta(w_\beta) = 90°$    (i) $\theta(w_\alpha) = 90°$, $\theta(w_\beta) = 90°$

Figure 7: Distributions $p(x \mid \alpha)$ for the linear-Gaussian generative map and single hyperplane classifier when $a = [1, \ 0]^T$. The orientation of $w_\alpha$ controls the direction in which the probability mass of $p(x \mid \alpha)$ shifts as $\alpha$ is varied, while the orientation of $w_\beta$ controls the rotation of each distribution $p(x \mid \alpha)$.

## B    Detailed analysis with linear-Gaussian generative map

In this section we provide empirical simulations supporting the analysis with a linear-Gaussian generative map in Section 4. Recall that we use the isotropic data distribution $X \sim \mathcal{N}(0, I)$, latent space prior $(\alpha, \beta) \sim \mathcal{N}(0, I)$, and

$$g(\alpha, \beta) = \begin{bmatrix} W_\alpha & W_\beta \end{bmatrix} \begin{bmatrix} \alpha \\ \beta \end{bmatrix} + \varepsilon,$$

where $W_\alpha \in \mathbb{R}^{N \times K}$, $W_\beta \in \mathbb{R}^{N \times L}$, and $\varepsilon \sim \mathcal{N}(0, \gamma I)$.

**Linear classifier.** Consider first the linear separator in $\mathbb{R}^2$ from Section 4, $p(Y = 1 \mid x) = \sigma(a^T x)$. With $K = L = 1$, learning an explanation entails learning the $w_\alpha, w_\beta \in \mathbb{R}^2$ that maximize the objective (3). As shown in Proposition 3, the data representation term $\mathcal{D}$ encourages $w_\alpha \perp w_\beta$; here we focus on the causal influence term $\mathcal{C}$. The decomposition in (4) shows that $\mathcal{C}$ depends on both $p(Y)$ and $p(Y \mid \alpha)$; Figure 7 visualizes how the distributions $p(Y \mid \alpha)$ change with $\alpha$ (gray ellipses) and $w_\alpha, w_\beta$ (subplots). Note first that the isotropy of $p(\alpha)$ means that $p(Y)$ has equal probability mass on either side of the classifier decision boundary, regardless of $w_\alpha$ and $w_\beta$. This implies that $H(Y)$ is invariant to $w_\alpha$ and $w_\beta$ for this classifier, a fact formalized in the proof of Proposition 3.

We next explore the role of $w_\alpha$ and $w_\beta$ in $p(x \mid \alpha)$ (and therefore $p(y \mid \alpha)$). Our causal objective $\mathcal{C}$ is large when the $p(y \mid \alpha)$ have low entropy in expectation over $\alpha$. Note from Figure 7 that $w_\alpha$ controls the direction in which the probability mass of $p(x \mid \alpha)$ shifts as $\alpha$ is varied, while $w_\beta$ controls the

Figure 8: Value of each causal objective variant in the linear-Gaussian generative map, linear classifier setting described in Section 4, as the orientations of $w_\alpha$ and $w_\beta$ are varied. The classifier decision boundary normal is $\theta(a) = 0°$. Each variant is maximized when $w_\alpha \propto a$ (i.e., $\theta(w_\alpha) = 0°$) and $w_\beta \perp a$ (i.e., $\theta(w_\beta) = 90°$). $\mathcal{C} = \mathcal{C}_{ju}$ refers to the causal objective (2) used in the main text.

Figure 9: Empirically-computed values of terms relevant to the causal objective variants in the linear-Gaussian generative map, "and" classifier setting described in Section 4. The angles of the classifier decision boundary normals are $\theta(a_1) = 0°$ and $\theta(a_2) = 90°$. Top row: log-likelihood used as $\mathcal{D}$; causal objective variants from Appendix A. $\mathcal{C} = \mathcal{C}_{ju}$ refers to the causal objective (2). Bottom row: terms in decomposition (4); "adjustment factors" from Proposition 4.

rotation of each distribution $p(x \mid \alpha)$. The causal objective $\mathcal{C}$ is maximized when the entropy of $p(y \mid \alpha)$ (in expectation over $\alpha$) is smallest — in other words, when the distributions $p(x \mid \alpha)$ have as little overlap possible with the classifier decision boundary. From Figure 7, we observe that this occurs when $w_\alpha$ is aligned with the decision boundary normal ($w_\alpha \propto a$) and when $w_\beta$ is orthogonal to the decision boundary normal ($w_\beta \perp a$). This selection of $w_\alpha$ and $w_\beta$ minimizes the range of $\alpha$ for which $p(x \mid \alpha)$ contains mass on both sides of the decision boundary.

Figure 8 shows the value of each of the causal objective variants described in Appendix A as the orientation of $w_\alpha$ and $w_\beta$ with respect to the classifier decision boundary normal $a$ are varied. For each combination of angles, we compute the causal objective using the sample-based estimate described in Appendix D with $N_\alpha = 2500$, $N_\beta = 500$, and the logistic sigmoid function $\sigma$ with steepness 5. (Note that in the training procedure we achieve satisfactory results with much lower $N_\alpha, N_\beta$.) These results verify the intuition presented above and formalized in Proposition 3: the causal effect is greatest when $w_\alpha \propto a$ and $w_\beta \perp a$. As noted in Section 4, in this setting both $\mathcal{C}$ and $\mathcal{D}$ encourage $w_\alpha$ and $w_\beta$ to be orthogonal.

**"And" classifier.** We now consider the "and" classifier in $\mathbb{R}^2$ from Section 4, $p(Y = 1 \mid x) = \sigma(a_1^T x) \cdot \sigma(a_2^T x)$, where we learn $K = 2$ causal explanatory factors and $L = 0$ noncausal factors. In this setting learning an explanation consists of learning $w_{\alpha_1}, w_{\alpha_2} \in \mathbb{R}^2$ maximizing (3).

Figure 9 shows how the value of the causal objective changes with the learned generative mapping in the linear-Gaussian setting of Section 4. The top row shows the terms in the objective (3): the likelihood and the causal objective variants described in Appendix A. The bottom row shows the components of these causal objective variants, which provide further intuition for their differences:

Figure 10: Empirically-computed value of combined objective (3) for the causal objective variants in the linear-Gaussian generative map, "and" classifier setting described in Section 4. The angles of the classifier decision boundary normals are $\theta(a_1) = 0°$ and $\theta(a_2) = 90°$. As $\lambda$ increases, the increased weight of the data representation term in the objective encourages the learned $w_{\alpha_1}$ and $w_{\alpha_2}$ to be more orthogonal to better represent the isotropic distribution of the data.

the first two plots show the decomposition of $\mathcal{C} = \mathcal{C}_{ju}$ from (4), and the remaining plots show the "adjustment factors" from Proposition 4 and Figure 6 that describe the differences between the causal influence objective variants. The logistic sigmoid with steepness 100 is used to implement the classifier, and the causal influence objective variants are computed with $N_\alpha = 2500$ and $N_\beta = 500$.

With the exception of the variant $\mathcal{C}_{iu}$, each of these causal objectives is maximized when $w_{\alpha_1}$ and $w_{\alpha_2}$ are aligned in the direction of maximum classifier change: $\theta(w_{\alpha_1}) = \theta(w_{\alpha_2})$ when $a_1 = [1,\ 0]^T$ and $a_2 = [0,\ 1]^T$ as in our example (see Figure 2(c-d)). Because with this classifier $\mathcal{C}$ does not encourage $w_{\alpha_1} \perp w_{\alpha_2}$, here the data representation term $\mathcal{D}$ serves to regularize $\mathcal{C}$. Figure 10 shows the value of the combined objective (3) for each causal influence variant and two different values of $\lambda$. We observe that as $\lambda$ increases and the weight of the data representation term increases, the optimal angles of $w_{\alpha_1}$ and $w_{\alpha_2}$ move in opposing directions from $45°$ (the angle of normal bisecting $a_1$ and $a_2$). This supports the intuition described in Section 4 and stylized in Figure 2(c-d).

## C  Proofs

### C.1  Proof of Proposition 2

Proposition 2 states that information flow coincides with mutual information in our DAG. Here we prove a generalization of the proposition that is also helpful when considering the conditional causal influence objective variants in Appendix A. Specifically, we consider the information flow from $U$ to $V$ imposing $W$:

**Definition 5** (Ay and Polani 2008 [7]). *Let $U$, $V$, and $W$ be disjoint subsets of nodes. The* information flow from $U$ to $V$ imposing $W$, *denoted $I(U \to V \mid W)$, is*

$$\mathbb{E}_{w \sim W} \left[ \int_U p(u \mid do(w)) \int_V p(v \mid do(u), do(w)) \log \frac{p(v \mid do(u), do(w))}{\int_{u'} p(u' \mid do(w)) p(v \mid do(u'), do(w))} dV \, dU \right],$$

*where $do(w)$ represents an intervention in a model that fixes $w$ to a specified value regardless of the values of its parents [6].*

**Proposition 6** (Information flow in our DAG). *The information flow from $\alpha$ to $Y$ imposing $\beta$ in the DAG of Figure 1(b) coincides with the mutual information of $\alpha$ and $Y$ conditioned on $\beta$,*

$$I(\alpha \to Y \mid do(\beta)) = I(\alpha; Y \mid \beta),$$

*where conditional mutual information is defined as $I(X; Y \mid Z) = \mathbb{E}_{X,Y,Z} \left[ \log \frac{p(x,y|z)}{p(x|z)p(y|z)} \right]$.*

*Proof.* The proof follows from the "action/observation exchange" rule of the *do*-calculus [6, Thm. 3.4.1]. This rule asserts that $p(y \mid do(x), do(z), w) = p(y \mid do(x), z, w)$ if $Y \perp Z \mid X, W$ in $\mathcal{G}_{\overline{X}\underline{Z}}$, the causal model modified to remove connections entering $X$ and leaving $Z$. When applied to our model, it yields

1. $p(Y \mid do(\alpha)) = p(Y \mid \alpha)$ (because $Y \perp \alpha$ in $\mathcal{G}_{\underline{\alpha}}$);

2. $p(\alpha \mid do(\beta)) = p(\alpha \mid \beta)$ (because $\alpha \perp \beta$ in $\mathcal{G}_{\underline{\beta}}$); and

3. $p(Y \mid do(\alpha), do(\beta)) = p(Y \mid \alpha, \beta)$ (because $Y \perp (\alpha, \beta)$ in $\mathcal{G}_{\underline{\alpha}, \underline{\beta}}$).

Starting with the definition of the information flow from $\alpha$ to $Y$ imposing $\beta$, we have that

$$\begin{aligned}
I(\alpha \to Y \mid do(\beta)) &= \mathbb{E}_\beta \left[ \int_\alpha p(\alpha \mid do(\beta)) \int_Y p(Y \mid do(\alpha), do(\beta)) \right. \\
&\qquad \left. \times \log \frac{p(Y \mid do(\alpha), do(\beta))}{\int_{a'} p(\alpha = a' \mid do(\beta)) p(Y \mid do(\alpha = a'), do(\beta))} \right] dY \, d\alpha \\
&= \mathbb{E}_\beta \left[ \int_\alpha p(\alpha \mid \beta) \int_Y p(Y \mid \alpha, \beta) \right. \\
&\qquad \left. \times \log \frac{p(Y \mid \alpha, \beta)}{\int_{a'} p(\alpha = a' \mid \beta) p(Y \mid \alpha = a', \beta)} \right] dY \, d\alpha \\
&= \mathbb{E}_\beta \left[ \int_{\alpha,Y} p(Y, \alpha \mid \beta) \log \frac{p(Y \mid \alpha, \beta)}{p(Y \mid \beta)} \right] dY \, d\alpha \\
&= \int_\beta p(\beta) \int_{\alpha,Y} p(Y, \alpha \mid \beta) \log \frac{p(Y \mid \alpha, \beta)}{p(Y \mid \beta)} dY \, d\alpha \, d\beta \\
&= \int_\beta p(\beta) \int_{\alpha,Y} p(Y, \alpha \mid \beta) \log \frac{p(Y \mid \alpha, \beta) p(\alpha \mid \beta)}{p(Y \mid \beta) p(\alpha \mid \beta)} dY \, d\alpha \, d\beta \\
&= \int_\beta p(\beta) \int_{\alpha,Y} p(Y, \alpha \mid \beta) \log \frac{p(Y, \alpha \mid \beta)}{p(Y \mid \beta) p(\alpha \mid \beta)} dY \, d\alpha \, d\beta \\
&= I(\alpha; Y \mid \beta).
\end{aligned}$$

$\square$

Proposition 2 follows from Proposition 6 by imposing the null set.

## C.2 Proof of Proposition 3

With $K = 1$ we can decompose $\mathcal{C}$ as

$$\mathcal{C} = I(Y; \alpha) = H(Y) - H(Y \mid \alpha). \tag{6}$$

where $H$ denotes entropy of a discrete random variable [83]. First consider the entropy term $H(Y)$. From the illustrations of $p(\widehat{X} \mid \alpha)$ in Figure 7, we can see in $\mathbb{R}^2$ that this entropy is constant for all values of $w_\alpha$ and $w_\beta$: regardless of their angle and offsets, the aggregate set of distributions $p(\widehat{X} \mid \alpha)$ is symmetric about the origin and so the probability mass of $p(\widehat{X})$ is spread symmetrically across both sides of the decision boundary. This idea is generalized in the following lemma, which shows that $H(Y)$ is equal to $\log(2) \approx 0.69$ nats for all values of $W$:

**Lemma 7.** *Under the conditions of Propsition 3, $H(Y) = \log(2)$ nats for all $W \in \mathbb{R}^{N \times N}$.*

*Proof.* Since $(\alpha, \beta) \sim \mathcal{N}(0, I)$, we have $\widehat{X} \sim \mathcal{N}(0, WW^T + \gamma I)$. Letting $U = a^T X$, we have $U \sim \mathcal{N}(0, a^T(WW^T + \gamma I)a)$ which we note has an even probability density function. Considering the classifier output probability marginalized over the generated inputs $\widehat{X}$, we have

$$\begin{aligned}
p(Y = 1) &= \mathbb{E}_{\widehat{X}}[p(Y = 1 \mid \widehat{X})] \\
&= \mathbb{E}_{\widehat{X}}[\sigma(a^T \widehat{X})] \\
&= \mathbb{E}_U[\sigma(U)] \\
&= \mathbb{E}_U[\sigma(U) - 0.5] + 0.5 \\
&\overset{(\star)}{=} 0.5
\end{aligned}$$

where in $(\star)$ we use the fact that since $U$ has an even probability density and $\sigma(U) - 0.5$ is an odd function, we have that $\mathbb{E}_U[\sigma(U) - 0.5] = 0$. Letting $h_b(p) = -(p \log p + (1 - p) \log(1 - p))$ denote the binary entropy function, we have that $H(\widehat{Y}) = h_b(p(\widehat{Y} = 1)) = h_b(0.5) = \log(2)$ nats. $\qquad\square$

We now consider the second term in (6), the conditional entropy $H(Y \mid \alpha)$. In $\mathbb{R}^2$ (Figure 7), this term corresponds to the average over $\alpha$ of the classification entropies for each distribution $p(\widehat{X} \mid \alpha)$ (depicted as individual ellipses). Intuitively, this entropy is small when many of the conditional distributions $p(\widehat{X} \mid \alpha)$ lie almost entirely on a single side of the decision boundary (corresponding to high classifier output agreement within each distribution, and therefore low entropy). The orientation of $w_\beta$ can reduce this term by rotating the data distributions so that their *minor*, not *major* axes cross the classifier, reducing the variance of classifier outputs in $\widehat{X} \mid \alpha$ for each unique $\alpha$. The orientation of $w_\alpha$ can reduce this term by moving the distributions $p(\widehat{X} \mid \alpha)$ away from the decision boundary (where disagreement in corresponding $Y$ values is lower) as quickly as possible as $|\alpha|$ increases.

**Lemma 8.** *Let $W = [w_\alpha \quad W_\beta]$, for $w_\alpha \in \mathbb{R}^N$ and $W_\beta \in \mathbb{R}^{N \times (N-1)}$. Suppose that each column $w_i$ of $W$ is bounded by $c > 0$, i.e., $\|w_i\|_2 \leq c$. Then under the conditions of Proposition 3, $H(Y \mid \alpha)$ is minimized when $w_\alpha = \pm c \frac{a}{\|a\|_2}$ and $W_\beta^T a = 0$.*

*Proof.* We have $\widehat{X} = w_\alpha \alpha + W_\beta \beta + \varepsilon$ with $\varepsilon \sim \mathcal{N}(0, \gamma I)$. For fixed $\alpha$, $p(\widehat{X} \mid \alpha) = \mathcal{N}(w_\alpha \alpha, W_\beta W_\beta^T + \gamma I)$. Defining $U = a^T X$, we have $U \mid \alpha \sim \mathcal{N}(\alpha a^T w_\alpha, a^T W_\beta W_\beta^T a + \gamma \|a\|_2^2)$. Then,

$$\begin{aligned}
p(\widehat{Y} = 1 \mid \alpha) &= \mathbb{E}_{\widehat{X}|\alpha}[p(\widehat{Y} = 1 \mid \widehat{X}, \alpha)] \\
&= \mathbb{E}_{\widehat{X}|\alpha}[p(\widehat{Y} = 1 \mid \widehat{X})] \\
&= \mathbb{E}_{\widehat{X}|\alpha}[\sigma(a^T X)] \\
&= \mathbb{E}_{U|\alpha}[\sigma(U)] \\
&\overset{(\star)}{=} \sigma\left(\frac{\alpha \langle a, w_\alpha \rangle}{\sqrt{1 + a^T W_\beta W_\beta^T a + \gamma \|a\|_2^2}}\right),
\end{aligned}$$

where $(\star)$ follows from the fact that for $Z \sim \mathcal{N}(\mu, \sigma^2)$, $\mathbb{E}_Z[\sigma(Z)] = \sigma\left(\frac{\mu}{\sqrt{1+\sigma^2}}\right)$.

We can now evaluate the entropy $H(Y \mid \alpha) = \mathbb{E}_{t \sim \alpha}[H(Y \mid \alpha = t)]$. Again denoting the binary entropy function by $h_b$, we have

$$H(Y \mid \alpha = t) = h_b(p(Y = 1 \mid \alpha = t))$$

$$= h_b(\sigma(s)) \quad \text{where } s := \frac{t \langle a, w_\alpha \rangle}{\sqrt{1 + a^T W_\beta W_\beta^T a + \gamma \|a\|_2^2}}$$

$$= h_b((\sigma(s) - 0.5) + 0.5).$$

Let $q := p - 0.5$ and define $\widetilde{h}_b(q) = h_b(q + 0.5)$ for $q \in [-0.5, 0.5]$ so that $\widetilde{h}_b$ is an even function. Therefore, $\widetilde{h}_b(q) = \widetilde{h}_b(|q|)$, and we have $h_b(p) = \widetilde{h}_b(p - 0.5) = \widetilde{h}_b(|p - 0.5|)$. Applying this fact yields

$$= \widetilde{h}_b(\sigma(s) - 0.5)$$

$$= \widetilde{h}_b(|\sigma(s) - 0.5|)$$

$$\stackrel{(\dagger)}{=} \widetilde{h}_b(|\sigma(|s|) - 0.5|)$$

where $(\dagger)$ follows since $|\sigma(s) - 0.5|$ is an even function of $s$. On $\mathbb{R}_{\geq 0}$ we have that $\widetilde{h}_b(\cdot)$ is a monotonically decreasing function and $|\sigma(\cdot) - 0.5|$ is a monotonically increasing function, and therefore $H(Y \mid \alpha = t) = \widetilde{h}_b(|\sigma(|s|) - 0.5|)$ is a monotonically decreasing function of $|s|$ where

$$|s| = \frac{|t| \, |\langle a, w_\alpha \rangle|}{\sqrt{1 + a^T W_\beta W_\beta^T a + \gamma \|a\|_2^2}}. \tag{7}$$

For any value of $t$, it is clear that the expression in (7) is maximized (and therefore $H(Y \mid \alpha = t)$ is minimized) with respect to $w_\alpha$ and $W_\beta$ when both $|\langle a, w_\alpha \rangle|$ is maximized and $a^T W_\beta W_\beta^T a$ is minimized. By the Cauchy-Schwarz inequality and from boundedness of the column magnitudes of $W$ by $c$, we have that $|\langle a, w_\alpha \rangle|$ is maximized at $w_\alpha = \pm c \frac{a}{\|a\|_2}$. Since $a^T W_\beta W_\beta^T a \geq 0$, this quadratic term is minimized at $W_\beta^T a = 0$ in which case $a^T W_\beta W_\beta^T a = 0$.

Since choosing $w_\alpha$ and $W_\beta$ in this way minimizes $H(Y \mid \alpha = t)$ for any $t$, we have that $H(Y \mid \alpha) = \mathbb{E}_{t \sim \alpha}[H(Y \mid \alpha = t)]$ is also minimized with this choice of $w_\alpha$ and $W_\beta$. $\qquad \square$

Since $H(Y)$ is constant for any $W$ (Lemma 7), we have that the conditions on $W$ described in Lemma 8 maximize $\mathcal{C} = I(\alpha; Y)$. We combine this result with the following lemma to characterize the minimum of the entire objective (3):

**Lemma 9.** *Suppose that $\varepsilon < 1$, $W \in \mathbb{R}^{N \times N}$, and that $X, \varepsilon, z \sim \mathcal{N}(0, I)$ in $\mathbb{R}^N$. With $U = Wz + \gamma \varepsilon$, $\mathrm{D}_{\mathrm{KL}}(p(X) \parallel p(U))$ is minimized by any orthogonal $W$ with columns normalized to magnitude $\sqrt{(1 - \gamma)}$.*

*Proof.* Noting that $U \sim \mathcal{N}(0, WW^T + \gamma I)$, we have from a standard result on KL divergence between multivariate normal distributions that

$$\underset{W}{\arg\min} \, \mathrm{D}_{\mathrm{KL}}(p(X) \parallel p(U)) = \underset{W}{\arg\min} \, \log \left| WW^T + \gamma I \right| + \mathrm{tr}((WW^T + \gamma I)^{-1}). \tag{8}$$

Since $WW^T + \gamma I$ is positive definite, there exists orthogonal $V$ and diagonal $\Lambda$ with positive entries $\{\lambda_i\}_{i=1}^N$ such that $WW^T + \gamma I = V \Lambda V^T$. We then have

$$\log \left| WW^T + \gamma I \right| + \mathrm{tr}((WW^T + \gamma I)^{-1}) = \log \left| V \Lambda V^T \right| + \mathrm{tr}((V \Lambda V^T)^{-1})$$

$$= \log |\Lambda| + \mathrm{tr}(\Lambda^{-1})$$

$$= \sum_i \log \lambda_i + \frac{1}{\lambda_i}. \tag{9}$$

(9) is minimized at $\lambda_i = 1$ for all $i$. Therefore, the minimizer of (8) is characterized by $WW^T = VV^T - \gamma I = (1 - \gamma)I$. Any orthogonal $W$ with column magnitudes equal to $\sqrt{1 - \gamma}$ satisfies this condition. $\qquad \square$

Combining these lemmas, consider the solution $w_\alpha = \sqrt{1-\gamma}\frac{a}{\|a\|_2}$, and $W_\beta$ with orthogonal, $\sqrt{1-\gamma}$-norm columns satisfying $W_\beta^T a = 0$. From Lemma 8 we have that this solution minimizes $H(Y \mid \alpha)$ within the class of $N \times N$ matrices whose column magnitudes are bounded by $\sqrt{1-\gamma}$. Combined with the invariance of $H(Y)$ to $W$ (Lemma 7), we have that $I(\alpha; Y)$ is maximized by this choice of $w_\alpha$ and $W_\beta$. From Lemma 9 we have that this solution also minimizes $\mathcal{D} = D_{\mathrm{KL}}(p(X) \| p(U))$, and thus this solution minimizes the objective (3) for any $\lambda > 0$.

### C.3 Proof of Proposition 4

Proposition 4 states the relationships between information flow-based objectives depicted graphically in Figure 6.

*Proof of (a).* We have that

$$I(Y; \alpha) = \frac{1}{K} \sum_{i=1}^{K} I(Y; \alpha_1, \ldots, \alpha_K)$$

$$= \frac{1}{K} \sum_{i=1}^{K} [I(Y; \alpha_i) + I(Y; \alpha_{\neg i} \mid \alpha_i)]$$

$$= \frac{1}{K} \sum_{i=1}^{K} I(Y; \alpha_i) + \frac{1}{K} \sum_{i=1}^{K} I(Y; \alpha_{\neg i} \mid \alpha_i).$$

$\square$

*Proof of (b).* First, note that

$$I(X; Y \mid Z, W) = \int_x \int_y \int_z \int_w p(x, y, z, w) \log \frac{p(x, y \mid z, w)}{p(x \mid z, w)p(y \mid z, w)} dx\,dy\,dz\,dw$$

$$= \int_x \int_y \int_z \int_w p(x, y, z, w) \log \frac{p(x, y, z, w)/p(z, w)}{p(x, z, w)/p(z, w)p(y, z, w)/p(z, w)} dx\,dy\,dz\,dw$$

$$= \int_x \int_y \int_z \int_w p(x, y, z, w) \log \frac{p(x, y, z, w)p(z, w)p(x, w)}{p(x, z, w)p(y, z, w)p(x, w)} dx\,dy\,dz\,dw$$

$$= \int_x \int_y \int_z \int_w p(x, y, z, w) \log \frac{p(x, y, z \mid w)p(z \mid w)p(x \mid w)}{p(x, z \mid w)p(y, z \mid w)p(x \mid w)} dx\,dy\,dz\,dw$$

$$= \int_x \int_y \int_z \int_w p(x, y, z, w) \left( \log \frac{p(x, y, z \mid w)}{p(x \mid w)p(y, z \mid w)} \right.$$

$$\left. - \log \frac{p(x, z \mid w)}{p(x \mid w)p(z \mid w)} \right) dx\,dy\,dz\,dw$$

$$= I(X; Y, Z \mid W) - \mathbb{E}_Y [I(X; Z \mid W)]$$

$$= I(X; Y, Z \mid W) - I(X; Z \mid W).$$

Applying this identity,

$$I(Y; \alpha \mid \beta) = \frac{1}{K} \sum_{i=1}^{K} I(Y; \alpha_i, \alpha_{\neg i} \mid \beta)$$

$$= \frac{1}{K} \sum_{i=1}^{K} [I(Y; \alpha_i \mid \alpha_{\neg i}, \beta) + I(Y; \alpha_{\neg i} \mid \beta)]$$

$$= \frac{1}{K} \sum_{i=1}^{K} I(Y; \alpha_i \mid \alpha_{\neg i}, \beta) + \frac{1}{K} \sum_{i=1}^{K} I(Y; \alpha_{\neg i} \mid \beta).$$

$\square$

*Proof of (c).* We have that

$$
\begin{aligned}
I(Y;\alpha \mid \beta) &= \int_Y \int_\alpha \int_\beta p(Y,\alpha,\beta) \log \frac{p(Y,\alpha \mid \beta)}{p(Y \mid \beta)p(\alpha \mid \beta)} dY d\alpha d\beta \\
&\overset{(\star)}{=} \int_Y \int_\alpha \int_\beta p(Y,\alpha,\beta) \log \frac{p(\beta \mid Y,\alpha)p(Y,\alpha)}{p(\beta)} \frac{p(\beta)}{p(\beta \mid Y)p(Y)} \frac{p(\beta)}{p(\beta \mid \alpha)p(\alpha)} dY d\alpha d\beta \\
&\overset{(\star\star)}{=} \int_Y \int_\alpha \int_\beta p(Y,\alpha,\beta) \log \frac{p(\beta \mid Y,\alpha)p(Y,\alpha)}{p(\beta)} \frac{p(\beta)}{p(\beta \mid Y)p(Y)} \frac{p(\beta)}{p(\beta)p(\alpha)} dY d\alpha d\beta \\
&= \int_Y \int_\alpha \int_\beta p(Y,\alpha,\beta) \log \frac{p(Y,\alpha)}{p(Y)p(\alpha)} \frac{p(\beta \mid Y,\alpha)}{p(\beta \mid Y)} dY d\alpha d\beta \\
&= \int_Y \int_\alpha \int_\beta p(Y,\alpha,\beta) \log \frac{p(Y,\alpha)}{p(Y)p(\alpha)} \frac{p(\beta \mid Y,\alpha)p(\alpha \mid Y)}{p(\beta \mid Y)p(\alpha \mid Y)} dY d\alpha d\beta \\
&= \int_Y \int_\alpha \int_\beta p(Y,\alpha,\beta) \log \frac{p(Y,\alpha)}{p(Y)p(\alpha)} \frac{p(\alpha,\beta \mid Y)}{p(\alpha \mid Y)p(\beta \mid Y)} dY d\alpha d\beta \\
&= \int_Y \int_\alpha \int_\beta p(Y,\alpha,\beta) \left( \log \frac{p(Y,\alpha)}{p(Y)p(\alpha)} + \log \frac{p(\alpha,\beta \mid Y)}{p(\alpha \mid Y)p(\beta \mid Y)} \right) dY d\alpha d\beta \\
&= I(Y;\alpha) + I(\alpha;\beta \mid Y),
\end{aligned}
$$

where $(\star)$ follows from Bayes' rule and $(\star\star)$ follows from the independence of $\alpha$ and $\beta$ in our model. $\square$

*Proof of (d).* Similar to (c). $\square$

---
**Algorithm 2** Sample-based estimate of $\mathcal{C}(\alpha; Y)$
___

**Input:** number of samples $N_\alpha$ and $N_\beta$, number of latent factors $K$ and $L$, number of classes $M$

   $I \leftarrow 0$
   $\boldsymbol{q}_y \leftarrow \text{zeros}(M)$
   **for** $i = 1$ **to** $N_\alpha$ **do**
     $\alpha \leftarrow K$-dimensional vector sampled from $\mathcal{N}(0, I)$
     $\boldsymbol{p}_{y|\alpha} \leftarrow \text{zeros}(M)$
     **for** $j = 1$ **to** $N_\beta$ **do**
       $\beta \leftarrow L$-dimensional vector sampled from $\mathcal{N}(0, I)$
       $x \leftarrow$ sample from $p(x \mid \alpha, \beta)$
       $\boldsymbol{p}_{y|\alpha} \leftarrow \boldsymbol{p}_{y|\alpha} + \frac{1}{N_\beta} p(y \mid x)$ (where $p(y \mid x) \in \mathbb{R}^M$ is the classifier probability for each class)
     **end for**
     $I \leftarrow I + \frac{1}{N_\alpha} \sum_{m=1}^{M} \boldsymbol{p}_{y|\alpha}[m] \log \boldsymbol{p}_{y|\alpha}[m]$
     $\boldsymbol{q}_y \leftarrow \boldsymbol{q}_y + \frac{1}{N_\alpha} \boldsymbol{p}_{y|\alpha}$
   **end for**
   $I \leftarrow I - \sum_{m=1}^{M} \boldsymbol{q}_y[m] \log \boldsymbol{q}_y[m]$
**Output:** $I$ (sample-based estimate of $I(\alpha; Y)$)
___

## D    Sample-based estimate of causal influence

Here we detail the sampling procedure for approximating the causal objective in (2). (The variants described in Appendix A can be approximated in similar fashion.) We have

$$\mathcal{C}(\alpha; Y) = I(\alpha; Y) = \int_\alpha p(\alpha) \left( \sum_y p(y \mid \alpha) \log p(y \mid \alpha) \right) d\alpha - \sum_y p(y) \log p(y)$$

where

$$p(y \mid \alpha) = \int_\beta \int_x p(y \mid x) p(x \mid \alpha, \beta) p(\beta) dx d\beta \tag{10}$$

and

$$p(y) = \int_{\alpha, \beta} \int_x p(y \mid x) p(x \mid \alpha, \beta) p(\alpha) p(\beta) dx d\alpha d\beta. \tag{11}$$

For fixed $\alpha$, we approximate (10) with $N_x$ and $N_\beta$ samples of $x$ and $\beta$, respectively, as

$$p(y \mid \alpha) \approx \frac{1}{N_\beta N_x} \sum_{j=1}^{N_\beta} \sum_{n=1}^{N_x} p(y \mid x^{(n)}),$$

where each $x^{(n)} \sim p(x \mid \alpha, \beta^{(j)})$ and $\beta^{(j)} \sim p(\beta)$. Similarly, we approximate (11) with $N_x$, $N_\alpha$, and $N_\beta$ samples of $x$, $\alpha$, and $\beta$, respectively, as

$$p(y) \approx \frac{1}{N_\alpha N_\beta N_x} \sum_{j=1}^{N_\beta} \sum_{i=1}^{N_\alpha} \sum_{n=1}^{N_x} p(y \mid x^{(n)}),$$

where each $x^{(n)} \sim p(x \mid \alpha^{(i)}, \beta^{(j)})$, $\alpha^{(i)} \sim p(\alpha)$, and $\beta^{(j)} \sim p(\beta)$. Therefore,

$$I(\alpha_i; y) \approx \frac{1}{N_\alpha N_\beta N_x} \left[ \sum_{i=1}^{N_\alpha} \sum_y \left( \sum_{j=1}^{N_\beta} \sum_{n=1}^{N_x} p(y \mid x^{(n)}) \right) \log \left( \frac{1}{N_\beta N_x} \sum_{j=1}^{N_\beta} \sum_{n=1}^{N_x} p(y \mid x^{(n)}) \right) \right.$$
$$\left. - \sum_y \left( \sum_{j=1}^{N_\beta} \sum_{i=1}^{N_\alpha} \sum_{n=1}^{N_x} p(y \mid x^{(n)}) \log \left( \frac{1}{N_\alpha N_\beta N_x} \sum_{j=1}^{N_\beta} \sum_{i=1}^{N_\alpha} \sum_{n=1}^{N_x} p(y \mid x^{(n)}) \right) \right) \right]$$

where each $x^{(n)} \sim p(x \mid \alpha^{(i)}, \beta^{(j)})$, $\alpha^{(i)} \sim p(\alpha)$, and $\beta^{(j)} \sim p(\beta)$.

The complete procedure is described algorithmically in Algorithm 2 with $N_x = 1$.

| Classifier Architecture |
|---|
| Input (28×28) |
| Conv2 (32 channels, 3×3 kernels, stride 1, pad 0) |
| ReLU |
| Conv2 (64 channels, 3×3 kernels, stride 1, pad 0) |
| ReLU |
| MaxPool (2×2 kernel) |
| Dropout ($p = 0.5$) |
| Linear (128 units) |
| ReLU |
| Dropout ($p = 0.5$) |
| Linear ($M$ units) |
| Softmax |

Table 1: Network architecture for MNIST Classifier

Figure 11: Partial details of parameter tuning procedure used to select $K$, $L$, and $\lambda$ for explaining MNIST 3/8 classifier using Algorithm 1. *Left:* In Step 1 we select the total number of latent factors $K + L$ needed to adequately represent the data distribution. *Center:* In Steps 2-3 we iteratively convert noncausal latent factors to causal latent factors until $\mathcal{C}$ plateaus. *Right:* After each increment of $K$, we adjust $\lambda$ to approximately achieve the value of $\mathcal{D}$ from Step 1.

# E  VAE experimental details and additional results

## E.1  Details and additional results for MNIST experiments

All experiments were run using a single Nvidia GeForce GTX 1080 GPU. The traditional MNIST training set was split into training and validation sets composed of the first 50,000 and remaining 10,000 images, respectively. The testing set was the same as the traditional MNIST testing set, composed of 10,000 images. These sets were down-selected to include only samples with the labels of interest. Input images were scaled so that the network inputs are in $[0, 1]^{28 \times 28}$.

The network architecture for the classifier used in the MNIST experiments is shown in Table 1 where $M$, the number of class outputs, varies depending on the classification task. The classifier was trained with a batch size of 64 and a stochastic gradient descent optimizer with momentum 0.5 and learning rate 0.1. The 3/8 classifier was trained for 20 epochs and the 1/4/9 classifier was trained for 30 epochs. The test accuracy of the classifier trained on both the 3/8 and 1/4/9 datasets was 99.6%.

The VAE architecture used to learn the generative map $g$ is shown in Table 2. The objective (3) was maximized with 8000 training steps, batch size 64, and learning rate $5 \times 10^{-4}$. At each training step, the causal influence term 2 was estimated using the sampling procedure in Appendix D with $N_\alpha = 100$ and $N_\beta = 25$. For experiments with digits 3 and 8, we selected $K = 1$, $L = 7$, and $\lambda = 0.05$ using the parameter selection procedure in Algorithm 1; Figure 11 shows intermediate results from this procedure.

| VAE Encoder Architecture | VAE Decoder Architecture |
|---|---|
| Input (28×28) | Input ($K + L$) |
| Conv2 (64 chan., 4×4 kernels, stride 2, pad 1) | Linear (3136 units) |
| ReLU | ReLU |
| Conv2 (64 chan., 4×4 kernels, stride 2, pad 1) | Conv2Transp (64 chan., 4×4 kernels, stride 1, pad 1) |
| ReLU | ReLU |
| Conv2 (64 chan., 4×4 kernels, stride 1, pad 0) | Conv2Transp (64 chan., 4×4 kernels, stride 2, pad 2) |
| ReLU | ReLU |
| Linear ($K + L$ units for both $\mu$ and $\sigma$) | Conv2Transp (1 chan., 4×4 kernel, stride 2, pad 1) |
| | Sigmoid |

Table 2: VAE network architecture used for MNIST and Fashion MNIST experiments.

Figure 12: Visualizations for learned latent factors for MNIST 3/8 classifier. Images in the center column of each grid are reconstructed samples from the validation set; moving left or right in each row shows $g(\alpha, \beta)$ as a single latent factor is varied. This plot shows the complete results from Figure 3; it includes sweeps for two additional samples and visualizations of all $L = 7$ noncausal factors.

Figure 12 shows additional results for the experiment of Figure 3, which visualizes the learned latent factors that explain the MNIST 3/8 classifier. Here we show latent factor sweeps from this experiment with additional data samples and all $K + L = 8$ latent factors.

Figure 13 shows an explanation of the same classifier architecture detailed in Table 1 trained on the MNIST digits 1, 4, and 9. We use the VAE architecture of Table 2 with $K = 2$ causal factors, $L = 2$ noncausal factors, and $\lambda = 0.1$, and estimated the causal influence portion of the objective using the sampling procedure in Appendix D with $N_\alpha = 75$ and $N_\beta = 25$. While the factor sweeps in Figure 13 provide a high-level indication of the data features each factor corresponds to, a practitioner may also wish to visualize the fine-grained transitions between each class. This can be achieved by sweeping each factor on a finer scale, as visualized by the zoomed in regions of Figure 13 as well as the more comprehensive sweeps in Figure 14.

Figure 13: Visualizations of learned latent factors for MNIST 1/4/9 classifier. Images in the center column of each grid are reconstructed samples from the validation set; moving left or right in each row shows $g(\alpha, \beta)$ as a single latent factor is varied. Varying the causal factors $\alpha_1$ and $\alpha_2$ control aspects that affect the classifier output (colored borders); varying the noncausal factors $\beta_1$ and $\beta_2$ affect only stylistic aspects such as rotation and thickness.

Figure 14: High-resolution transition regions of the first causal factor in explaining the MNIST 1/4/9 classifier. Visualizing high-resolution latent factor sweeps can allow a practitioner to more easily identify which data features correspond to each underlying factor. For example, one can observe in the second row from the bottom how increasing $\alpha_1$ causes the left branch of the digit '4' to smoothly transition into completing the loop of the digit '9' while the digit stem remains fixed.

### E.2 Details and additional results for comparison experiments

Figure 4 compares our latent factor-based local explanations to the local explanations of four popular explanation methods. We generate explanations of the same CNN classifier trained on MNIST 3 and 8 digits described in Appendix E.1. The data samples explained in Figure 4 are the first example of each class in the MNIST validation set.

**Implementation details of other methods.** The following procedures were used to generate the results for LIME, DeepSHAP, IG, and L2X shown in Figure 4 (left):

- **LIME [17].** The LIME framework trains a sparse linear model using superpixel features. Following the recommendation in the authors' code, we generate superpixels using the Quickshift segmentation algorithm from scikit-image with kernel size 1, maximum distance 200, and color/image-space proximity ratio 0 (as the MNIST digits are grayscale). The LIME local approximation is fit using the default kernel width of 0.25, 10,000 samples, and $K = 10$ features. Figure 4 show superpixels identified as contributing positively (red) and or negatively (blue) to the classification decision.

- **DeepSHAP [25].** The DeepSHAP method uses the structure of the classifier network to efficiently approximate Shapley values, a game-theoretic formulation for how to optimally distribute rewards to players of a cooperative game. The Shapley values displayed in Figure

Figure 15: Complete results for local explanations of '3' and '8' from Figure 4. Our explanations are able to differentiate causal aspects (pixels that define 3 from 8) from purely stylistic aspects. Only the causal factor $\alpha$ controls changes in data space that result in a change in classifier output.

4 can be interpreted as the (averaged) importance of each pixel for explaining the difference between $f(x)$ and $\mathbb{E}_{x \sim X}[f(x)]$. We train the explanation model using 1000 randomly chosen samples from the training set. The DeepSHAP method produces explanations for each possible class; we display the Shapley values corresponding to the classifier class (i.e., the top image shows the explanation for ground truth class 3 and the bottom image shows the explanation for ground truth class 8).

- **IG [24].** The integrated gradients (IG) method integrates the gradient of the classifier probabilities with respect to the input as the input changes from a "baseline." We use an all-zero image as the baseline and the trapezoid rule with 50 steps to approximate the integral. The output in Figure 4 shows the integrated gradient explanation for each input image.

- **L2X [11].** The learning to explain (L2X) algorithm learns a mask of features $S$ that (approximately) maximizes $I(Y; X \odot S)$. Following [11, Sec. 4.3], we find a mask with $k = 4$ active superpixels, each of size $4 \times 4$. The neural network parameterizing the "explainer" model $p(S \mid X)$ consists of two convolutional layers (32 filters of size $2 \times 2$ each with relu activation, each followed by a max pooling layer with a $2 \times 2$ pool size), followed by a single $2 \times 2$ convolutional filter. This explainer network learns a $7 \times 7$ mask, with each element corresponding to a $2 \times 2$ superpixel in data space. The neural network parameterizing the variational bound $q(Y \mid X \odot S)$ consists of two convolutional layers, each containing 32 filters of size $2 \times 2$, using relu activation, and followed by a max pooling layer with $2 \times 2$ pool size; followed by a dense layer. The networks parameterizing $p(S \mid X)$ and $q(Y \mid X \odot S)$ were trained together with 10 epochs of the 9943 MNIST training samples of 3's and 8's and the outputs $Y$ of the convolutional neural network classifier described in Appendix E.1.

**Complete results for our method.** In Figure 4 (right) we show only latent factor sweeps for the the causal factor $\alpha$ and a single noncausal factor $\beta_7$. Figure 15 shows complete local explanations with each noncausal factor. Our explanations use the VAE framework described in Appendix E.1.

### E.3  Details and additional results for fashion MNIST experiments

Our training set was the same as the traditional Fashion MNIST training set, composed of 60,000 images. The Fashion MNIST testing set was split into validation and testing sets composed of the first 6,000 and last 4,000 images, respectively. These sets were down-selected to include only samples with the labels of interest — in our experiment, classes 0 ('t-shirt/top'), 3 ('dress'), and 4 ('coat'). Input images were scaled so that the input images were in $[0, 1]^{28 \times 28}$.

Figure 16: Partial details of parameter tuning procedure used to select $K$, $L$, and $\lambda$ for explaining a classifier trained on classes 0, 3, and 4 of the fashion MNIST dataset using Algorithm 1. *Left:* in Step 1 we select the total number of latent factors $K + L$ needed to adequately represent the data distribution. *Center:* In Steps 2-3 we iteratively convert noncausal latent factors to causal latent factors until $\mathcal{C}$ (shown in nats) plateaus. *Right:* After each increment of $K$, we adjust $\lambda$ to approximately achieve the value of $\mathcal{D}$ from Step 1.

The same classifier architecture described in Table 1 was used in this experiment. The classifier was trained with 50 epochs, a batch size of 64, a stochastic gradient descent optimizer with momentum 0.5 and learning rate 0.1. Because the classes used ('t-shirt/top,' 'dress,' and 'coat') are similar, this classifier task is more challenging than the MNIST digit classification task; the test accuracy of the classifier was 95.2%.

The same VAE architecture described in Table 2 was used to learn the generative map $g$. The objective (3) was maximized with 8000 training steps, batch size 32, and learning rate $10^{-4}$. At each training step, the causal influence term (2) was estimated using the sampling procedure in Appendix D with $N_\alpha = 100$ and $N_\beta = 25$. Using the parameter selection procedure in Algorithm 1, we selected $K = 2$, $L = 4$, and $\lambda = 0.05$; Figure 16 shows intermediate results from this procedure.

Figure 17 contains the complete results from the experiment in Figure 5 (right), showing a complete visualization of the global explanation learned for this classifier.

Figure 17: Visualizations of learned latent factors for Fashion MNIST classifier trained on classes 't-shirt-top,' 'dress,' and 'coat.' Images in the center column column of each grid are reconstructed samples from the validation set; moving left or right in each row shows $g(\alpha, \beta)$ as a single latent factor is varied. This plot shows the complete results from Figure 5 (right); it includes sweeps for two additional samples and visualizations of all $K + L = 6$ latent factors.

# F    Selecting generative model capacity

One practical decision to make when constructing explanations using our method is selecting the *capacity* of the generative model $g$. Set too low, the generative model will have insufficient capacity to represent the data distribution and classifier, reducing the quality of the explanation. Set too high, the generative model will require a more time- and energy-intensive training procedure.

We can use results from [75] to bound the capacity mismatch of our explainer (i.e., explainer error in predicting classifier outputs) with the $I(\alpha; Y)$ part of our objective. In practice, this result means that a sufficiently large value of $I(\alpha; Y)$ serves as a certificate that the explainer complexity is sufficient to explain the classifier. Below, we show details of this analysis and empirically demonstrate how $I(\alpha; Y)$ can be used to select an architecture with sufficient capacity.

## F.1    $I(\alpha; Y)$ serves as a certificate of sufficient explainer capacity

One reasonable measure for the quality of an explanation method is how accurately the black-box's classifications can be predicted from the explanation alone. If this prediction is accurate, then in a predictive sense the explanation has captured the relevant information about the classifier's behavior. In our model, the estimator that minimizes prediction error is the MAP estimate of the classifier's output from $p(Y \mid \alpha)$, where $p(Y \mid \alpha)$ is determined by marginalizing $p(Y \mid X)p(X \mid \alpha, \beta)p(\beta)$ over $\beta$ and $X$. As we show below, we can upper bound the error of this MAP estimator *directly* by the causal effect $I(\alpha; Y)$ of $\alpha$ on $Y$, the quantity our method explicitly optimizes.

Specifically, let $\pi(Y \mid \alpha) \coloneqq \int_{\alpha} [1 - \max_y p(y \mid \alpha)] p(\alpha) d\alpha$ denote the expected error of this MAP estimator, averaged over the prior distribution on causal factor $\alpha$. From [75], we have

$$\phi^*(\pi(Y \mid \alpha)) \leq H(Y \mid \alpha),$$

where $H(Y \mid \alpha)$ is the conditional entropy of $Y$ given $\alpha$, and $\phi^*$ is a monotonically increasing, invertible function. Define $\widetilde{\phi} = (\phi^*)^{-1}$. Since $H(Y \mid \alpha) = H(Y) - I(Y; \alpha) \leq \log M - I(Y; \alpha)$ [83], we have

$$\pi(Y \mid \alpha) \leq \widetilde{\phi}(\log_2 M - I(Y; \alpha)) \tag{12}$$

where $I(Y; \alpha)$ is measured in bits.

If we take the prediction error of $Y$ from $\alpha$ as a measure of "mismatch" between our trained model and the blackbox classifier, (12) bounds this mismatch by the causal effect term in our objective and can serve as a certificate for having sufficient network capacity. For example, in 3-class Fashion MNIST ($M = 3$), a value of $I(\alpha; Y) = 1.03$ nats as in Figure 16 results in a bound of $\pi(Y \mid \alpha) \leq 0.05$. This translates to a MAP estimator of $Y$ from $\alpha$ having a black-box output prediction error of less than $5\%$, or that the causal factors can explain at least $95\%$ of the black-box's behavior. If this prediction accuracy is satisfactory, then the capacity of the generator $g$ is sufficient to learn appropriate latent factors and their mapping to the data space. If this prediction accuracy is not satisfactory, a class $G$ of generative models $g$ with higher capacity can be used. This will provide the model with more flexibility to optimize $I(\alpha; Y)$ and reduce prediction error.

## F.2    Empirical results

The drawback of a VAE with insufficient capacity can be seen in Figure 18, which shows the causal effect and data fidelity terms of the objective (3) as the VAE capacity and tuning parameter $\lambda$ are modified. The VAE in each trial, which is applied to explain the 3-class Fashion-MNIST classifier considered in the quantitative experiments of Section 5, uses the architecture described in Table 2 with $K = 2$ and $L = 4$ but with a variable number of convolutional filters in each layer of the encoder and decoder (see Table 3). The values of $\mathcal{C}$ and $\mathcal{D}$ reported in Figure 18 are the average values in the last 50 training steps for each model. The dotted line in Figure 18 represents the maximum achievable value of $I(\alpha; Y)$ in this three class setting, $\log(3) \approx 1.1$ nats.

As discussed in Section 3.4, the tuning parameter $\lambda$ dictates the trade-off between the objective's causal effect term $\mathcal{C}$ and data fidelity term $\mathcal{D}$. When the number of filters per layer is small, however, the model has insufficient capacity to simultaneously achieve a satisfactory value of both $\mathcal{C}$ and $\mathcal{D}$.

Figure 19 shows partial resulting explanations generated by an explainer with insufficient capacity (8 filters per convolutional layer; Figure 19(a–b)). Although the causal and noncausal factors do

Figure 18: Post-training value of the (a) causal effect and (b) data fidelity terms in the objective (3) for various capacities of VAE. The capacity is modified by changing the number of convolutional filters in each layer.

| Filters per convolutional layer | Encoder parameters | Decoder parameters |
|:---:|:---:|:---:|
| 8 | 6,916 | 4,937 |
| 16 | 17,916 | 13,969 |
| 32 | 52,204 | 44,321 |
| 48 | 102,876 | 91,057 |
| 64 | 169,932 | 154,177 |

Table 3: Number of VAE parameters when $K + L = 6$.

indeed roughly correspond to classifier-relevant and classifier-irrelevant data aspects in the sense that changing $\alpha_1$, but not $\beta_1$, produces changes in the classifier output, the effect of the model's limited ability to represent the data distribution is evident in the weak correspondence of the generated samples to training samples. Meanwhile, the same explanation generated by an explainer with sufficient capacity (64 filters per convolutional layer; Figure 19(c–d)) shows both effectively disentangled classifier-relevant/irrelevant data aspects and generated samples that appear to lie in the training data distribution.

Figure 19: Global explanations with $\lambda \approx 0.013$ and varying VAE model capacity. (a–b) 8 filters per convolutional layer, defining a VAE with insufficient capacity to represent the data distribution. (c–d) 64 filters per convolutional layer, defining a VAE with sufficient capacity to represent the data distribution.