[Reviews · NeurIPS 2020]

Review 1

Summary and Contributions: This paper presents a generative model to "explain" any given black-box classifier and its training dataset. By "explain", the authors mean that a hidden factor can be discovered to control or intervene in the output of the classifier. The discovery is based on a proposed maximization objective, which consists of two terms: 1) a proposed Information Flow that denotes the causal effect from the hidden factor to the classifier output and 2) a distribution similarity to impose the discovered hidden factor can generate back the feature space.

Strengths: 1. well-written. I believe that any reader of broad interest can understand the story. 2. simple yet effective method 3. extensive ablative experiments.

Weaknesses: 1. The authors unnecessarily frame the method into causal inference. This can be clearly seen in Fig1(b), where the hidden variable of interest: \alpha and \beta are all root nodes, and there are no mediation nodes. Therefore, this renders almost all techniques of causal inference such as do-operation and counterfactuals trivial. Though the authors claim that the unawareness of causal knowledge is a benefit, I suggest the authors change a storyline. 2. Though the discovered \alpha shows the effective interventional effect on the classifier output, it is still un-nameable and coarse-grained, that is to say, any user is still hard to explain the decision of the classifier. It seems that the authors over-claim that \alpha is a more "expressive vocabulary than feature selection" in Line 41 to 45. 3. The mechanism for imposing \beta to cause zero-effect on Y is proved in Proposition 3, which is based on the assumption of Gaussian distribution. However, this assumption is too strong. In fact, for more realistic classifiers, the data is not Gaussian. Therefore, in general, there is no guarantee for the zero-effect of \beta on Y. Correct me if I were wrong. 4. Another downside of \alpha is that it requires the use of the training data. I think it is not practical as other feature selection method, who only requires the trained feature extractor (backbone model). As you know, in a practical scenario, training data is confidential. 5. This paper is essentially about "disentanglement". I suggest the authors compare and discuss the following related work (but not limited to): R1: Counterfactuals uncover the modular structure of deep generative models. R2: Challenging Common Assumptions in the Unsupervised Learning of Disentangled Representations R3: Robustly Disentangled Causal Mechanisms: Validating Deep Representations for Interventional Robustness R4: Towards a Definition of Disentangled Representations

Correctness: 1. Some of the sentences in Introduction needs more concrete examples or more precise causal language, such as "A central challenge" in line 24. 2. The causal graph may involve (unobserved) confounders, to make your causal graph more reasonable. Or, you may want to explain more about the validity of the proposed (though its simple) causal graph.

Clarity: Yes.

Relation to Prior Work: See the above weakness.

Reproducibility: Yes

Additional Feedback: See the above weakness. [Update] I appreciate the authors' feedback, especially for their plan to revise the storyline of the paper. However, I am still puzzled by their clarification on the use of classifiers' training data. I'm delighted to raise the score from 5 to 6.


Review 2

Summary and Contributions: To resolve the two problems: generating counterfactual instances within data distributions and measuring the causal influence of data aspects accurately, the authors propose a framework with two components to address the first problem and a metric for the second problem.

Strengths: This work proposes to learn low-dimensional latent variables and generate modified samples for explaining the behavior of the classifier. The proposed method is generative and model agnostic. Experiments are done across multiple datasets and settings to show the effectiveness of the method.

Weaknesses: Using mutual information as the measure of causal influence can be problematic [1]. The difference between generative models that learn disentangled representations and the method proposed by this work should be mentioned (e.g., [2]), because if a model learns disentangled representations and can generate good samples then it has the same functionality as the proposed method. It would also be much more interesting if the authors could compare their method with such generative models instead of comparing with those feature selection or salience mapping methods. I guess the difference is that the authors explicitly extract \alpha (causal factors) and \beta (non-causal factors). [1] Chang, Shiyu, Yang Zhang, Mo Yu, and Tommi S. Jaakkola. "Invariant rationalization." ICML 2020. [2] Higgins, Irina, Loic Matthey, Arka Pal, Christopher Burgess, Xavier Glorot, Matthew Botvinick, Shakir Mohamed, and Alexander Lerchner. "beta-vae: Learning basic visual concepts with a constrained variational framework." (2016).

Correctness: It is questionable whether mutual information is a good metric for causal influence. As we know MI measures a type of correlation. The comparison with baselines in the VAE experiment is also not really a comparison as the explanation generated by the baseline methods are not in the same format as the proposed method. In figure 5, why removing \alpha_2 (higher information flow) has less effect than \alpha_1 (lower information flow)?

Clarity: The paper is well written. Methodologies and experiments are well described and easy to be understood.

Relation to Prior Work: Compared to the most related previous work that is causal and generative, and construct samples from latent factors, the authors clearly describe their method: (1) does not require labeled concept, (2) does not suffer from the limitation of the definition of ACE, and (3) is model agonostic.

Reproducibility: Yes

Additional Feedback: Updates: I think the paper is acceptable but not outstanding, so I maintain my score. Some of my concerns are not addressed in the authors' feedback. First, although the authors mentioned two reference to support their point that MI can measure causal influence. The fact is that you can never measure causality with a statistical association metric. Second, I mentioned "The comparison with baselines in the VAE experiment is also not really a comparison as the explanation generated by the baseline methods are not in the same format as the proposed method." While it seems not a concern for the authors. Maybe this kind of comparison is the usual case in the interpretability literature.


Review 3

Summary and Contributions: This is a very well-written (albeit simple) paper on constructing post-hoc explanations for classifiers. The idea is to learn a generative model over inputs, i.e., X = g(alpha,beta), where alpha and beta are random variables, in a manner where only alpha is "causally" related to the output of the classifier operating on X. The separation between alpha and beta is learned by balancing two criteria for the generative model. First, one maximizes the information between alpha and the classifier output (via X), and, second, one makes the generative model accurately capture the data distribution over X. Since the causal model is a simple chain, {alpha,beta} -> X -> Y, where the last step is the classifier to be explained, any causal arguments relating to alpha's influence on Y in terms of do-calculus naturally immediately collapse into information arguments. This is shown separately as a "result" but is perhaps more like an observation. The approach is illustrated on simple problems, from an illustrative synthetic case (linear classifier + Gaussian over X) to MNIST (8's vs 3's) and fashion MNIST datasets.

Strengths: The paper is very clean and well-written, and the illustrations provided are helpful and insightful (albeit simple).

Weaknesses: The approach is quite simple and illustrated across relatively simple problems (linear + Gaussian, MNIST 8 vs 3, and fashion MNIST). The theoretical results, propositions 2 and maybe also 3, are almost immediate observations rather than "surprising" results. The authors' feedback does not change the opinion in this respect. I believe the key strength of the paper is its clarify of presentation rather than the results. It's not clear how the method would scale to problems where the classifier is complex and derives its output from many causally relevant factors (not just 1-3), or where the distribution over the inputs is hard to approximate (often in real problems). Figure 4: it seems a bit strange to compare explanations that are either saliency maps or actual samples. Samples are good in simple cases but are unlikely to be useful in more complex scenarios where one doesn't already know the key differentiating dimensions.

Correctness: seem correct.

Clarity: one of the key strengths of the paper is clear presentation.

Relation to Prior Work: The use of mutual information to guide latent representations has been proposed before, e.g., InfoGAN: Interpretable Representation Learning by Information Maximizing Generative Adversarial Nets https://arxiv.org/abs/1606.03657 But absent the classifier from X to Y. It would also be helpful to relate the approach to learning disentangled representations, e.g., Disentangling by Factorising https://arxiv.org/pdf/1802.05983.pdf

Reproducibility: Yes

Additional Feedback:


Review 4

Summary and Contributions: The authors provide a framework for generating explanations for a black-box classifier by inferring low-dimensional latent factors informative for the classifier decisions. The authors provide theoretical justification for their approach based on the information-theoretic measures of causality. Furthermore, they illustrate and analyse the behaviour of their method first on a simple illustrative example, and then on a real-world application.

Strengths: The paper targets a highly relevant and important problem. The authors propose a new, original angle of looking at explainability, and skillfully implement their idea, providing both a theoretical justification and a number of insightful illustrations. It is very likely that this approach will be useful in many applications, benefitting a large number of ML community members. Lastly, the paper is very well written. It is truly a pleasure to read. It seems that a lot of thought was put into the presentation: great phrasing, well thought out illustrative examples, and thoroughly explained intuitions in the supplementary materials together create a very favourable impression.

Weaknesses: Although strong overall, the paper has noticeable imperfections. Firstly, I have ambivalent feelings about a number of presentational decisions, which I describe in the “Clarity” section. Additionally, in the “relation to prior work” section I give my reasons for why I believe that some improvements are very much desired when it comes to placing the work in the broader context. Lastly, the absence of analysis / discussion about practical applicability of the proposed methods partially limits potential applications. In particular, it seems that in order to apply this model to large-scale image classification problems (e.g. ImageNet), one would need to train a network analogous in capacity to the black-box classifier that has to be explained. It may become a very serious practical concern in many settings. It would be good to discuss potential workarounds (e.g. using a pre-trained classifier, etc.). Moreover, since the black-box classifier can, potentially, have an unknown architecture, it seems important to estimate the consequences of using a model that differs in capacity from the black-box that has to be explained. I believe that overall, the contribution is still above the acceptance threshold, but only marginally. It can become a truly exceptional paper if these issues are fully addressed.

Correctness: I believe that the paper is well thought-out methodologically, and that the approach is, overall, sound. While the authors don't conduct extensive experiments on large-scale problems, they compensate that by a number of great illustrative examples which help to both test and illustrate the method's performance. I think that overall, such a detailed and thoughtful approach presents a great example of high-quality research.

Clarity: The paper is very well written and is a pleasure to read. I have, however, a few concerns about the choice of the overall paper structure and certain presentation decisions. I must first mention that they are highly subjective and may be partially a matter of taste. As such, they did not strongly impact my assessment, but I want to mention it to a) let authors know that there may be an issue b) communicate these considerations to other reviewers. The first concern is the author’s decision of centering the paper presentation around the theme of causal explanations, while in the model that the authors consider, the causal part is equivalent to maximizing mutual information between a subset of latent features and the classifier decision. The connection is insightful, but the problem is that the notion of causality is often overused and overloaded with different meanings (the authors themselves have to right away clarify what they mean in the abstract). When speaking about classification, a common false expectation may be that readers may assume that explanation works by uncovering causal connections between features, not between the feature and the class. Moreover, the commonsense explanation given in the abstract “the explanation is causal in the sense that changing learned latent factors produces a change in the classifier output statistic” does not allow to intuitively distinguish between the proposed approach and that of many other post-hoc explainability methods. When viewed in this light, it may have been beneficial to refer to the explanations/features as “informative”, while still keeping the connection to causal explanations in the paper. An additional benefit of doing so would be that it automatically helps to draw a connection with works on VAEs and InfoGAN. This connection seems to be there, but it was never discussed in detail (I speak more about that in the “relation to prior work” section). The second, related, but separate concern is that the authors seem to be slightly downplaying the role and depth of certain previous contributions. Thus, they mention, for example, that before the authors derived the mutual information objective “from first principles of causality”, it was used as a “heuristic” in a number of previous works (e.g. in Variational Information Maximization for Feature Selection, 2016). I believe that the idea of maximizing mutual information may be seen by many as a valid start of a theoretical justification for a method. While it is always nice to re-interpret the objective in a new light, providing additional depth, I think that it may be better to frame the theoretical connection as a new interpretation, not as a fist justification of a heuristic that was hitherto used blindly. Lastly, the paper does not address the limitations of the method (i.e. potentially heavy demands for computational power, the need to appropriately choose the right model architecture for the problem at hand). The paper is potentially very useful for ML practitioners and industry applications, hence it seems especially important. Overall, I feel that these (essentially stylistic) decisions may make it more difficult for members of the community to correctly evaluate the merits and limitations of the contribution both on the practical and on the theoretical sides.

Relation to Prior Work: The authors give a very good overview of related methods in the domain of explainable AI. In my opinion, it may be, however, beneficial to discuss in more detail how the proposed method fits into a broader context of work on generative models. The proposed model optimizes mutual information between the class (predicted by the black-box model) and the learned features and involves distance measure between the data and generator distribution D(p(g(alpha, beta)), p(X)). This places the model close to the works on VAEs, InfoGANs, and related generative models. In fact, if we cut the “black box classifier” part of the Figure 1a), we get a supervised method for discovering meaningful features, very similar in spirit to InfoGAN (although InfoGAN is marketed as an unsupervised method, it is possible to introduce conditioning). An even closer connection is to to the work “Multi-Level Variational Autoencoder: Learning Disentangled Representations from Grouped Observations” (2018) which proposes a method of discovering disentangled “style” and “content” features when provided with class (group) information. If we provide classifier predictions as group information, we get an alternative approach to what is done in the present contribution (although such a method would be more crude in that the group attributions are, by default, not weighted). Overall, I believe that it is very important to discuss these and related connections, for two reasons. Firstly, it may be that the proposed method has much broader applicability outside the scope of explainable AI. On the other hand, the absence of such discussion makes it more difficult to evaluate the novelty and originality of the present contribution.

Reproducibility: Yes

Additional Feedback: I am personally very torn about my decision for this paper. I believe that at present, the paper is already strong, and it is impossible for me to not recommend it for acceptance. At the same time, the issues I described do the contribution a bad favor in that overall, now it leaves a slightly mixed impression, while it seems to have the potential to unequivocally and uniformly produce a stellar impression. I almost wish it were rejected so that authors could address the issues and turn the paper into a gem it could be. This line of reasoning, of course, can not justify me not recommending an acceptance. Overall, I evaluate the paper as is, and I believe that it is, at present, slightly above the acceptance threshold, despite the issues I described. Suggestions: It may be very interesting to see how the model behaves depending on the model capacity differences between the black box classifier and the explainer model. For example, do the explanations remain meaningful when the explainer model is much smaller than the classifier itself. Does it behave meaningfully if its capacity is, on the contrary, much higher, etc. Practically, it is one of the most concerning issues. Since the method is supposed to work with a black box classifier, we can never know whether we undershoot or overshoot in our model capacity. ## I have read the author's response. Many of my concerns were addressed. I am especially happy to hear about the changes to the storyline. I am happy to increase my score from 6 to 7.

[Author Response · NeurIPS 2020]

We thank the reviewers for their substantive and constructive feedback, and appreciate their assessment of our explanation
framework as "simple yet effective" **(R1)** with convincing experiments and clear presentation. We particularly appreciate
**(R4)**'s comment that our "detailed and thoughtful approach presents a great example of high-quality reserch." The main
reviewer concerns were narrative-related; we have addressed these concerns (and other specific suggestions) in revision.

**1: Technical misconceptions in reviews.**

• **(R1)** (Weakness 3) Section 4 illustrates and tests the performance of our method in a controlled setting; here we
define the data distribution to be Gaussian, but outside of this section we do *not* make any assumptions on the data
distribution. Also, our objective does *not* attempt to minimize $I(\beta; \widehat{Y})$ (but see Appendix A for discussion of potential
objective variants).

• **(R1)** (Weakness 4) We do *not* require the training data originally used to train the classifier, which we agree would be
problematic. Our explanations are based on a data distribution, but this need not be the training distribution.

• **(R1) (R2)** The validity of our DAG stems from the independence of $(\alpha, \beta)$, which we impose with an isotropic normal
prior. (We have clarified this point in Sections 3.3, 4, and 5). Although MI is indeed a correlative metric in general,
the independence of $(\alpha, \beta)$ allows us to show that in our framework it quantifies information flow, a well-founded
node-based metric for causal influence (see refs 7, 55). This argument is indeed very simple (as pointed out by
**(R1) (R2)**), but it depends crucially on our modeling decisions. Note that in the generative modeling literature the
assumption of latent factor independence is common. One might alternately allow dependencies between the latent
factors, jointly learning their causal structure with the generative network. However, without labeled side-information
that would give these features semantic meaning, in our view our independence-by-construction approach is more
useful to generate parsimonious explanations.

**2: Changes to storyline.**  The most consistent reviewer concerns were narrative-related: (1) **(R1) (R2) (R4)** com-
mented that the role of causality in our framework was overstated or unnecessary (since by enforcing the $(\alpha, \beta)$ to be
independent our metric for causal influence simply reduces to MI), and (2) **(R1) (R2) (R3) (R4)** commented that our
narrative did not engage with the disentangled representation literature. We thank the reviewers for these suggestions
and have addressed them with careful changes to our storyline and a major addition to our related work section:

• As suggested by **(R1) (R4)**, we have adjusted our storyline to change the perspective from which we approach
causality, instead highlighting the aspects of the model that allow the simple MI metric to be interpreted causally
**(R2)**. We also clarified that the prior enforcing independence of $(\alpha, \beta)$ leads to the validity of our DAG **(R1)**. We
agree with **(R4)** (and related work) that MI on its own can serve as a valid theoretical justification for explanation,
and have reworked our discussions of MI-based methods to remove the impression of downplaying, emphasizing
instead that our framework provides the *complementary* benefits of causal and information-theoretic interpretations.

• We believe that the disentanglement framing suggested by **(R1) (R2) (R3) (R4)** is exciting and could open new
research directions. Our method can indeed be thought of as an information-based disentanglement procedure. Unlike
techniques such as InfoGAN, our method (1) uses a classifier as side information; (2) separates classifier-relevant
from classifier-irrelevant features, making the framework suitable for explanation; and (3) allows the MI metric to be
interpreted as a measure of causal influence of disentangled features on the classifier output. We have adjusted the
storyline throughout the paper to add this perspective, and have added a paragraph in the background to discuss the
relation to specific disentanglement techniques (including all suggestions from **(R1) (R2) (R3) (R4)**) in more detail.

**3: Additional experiments and discussion.**

• **(R4)** We agree that computational cost is a concern. We have added a sentence in the main text mentioning this
cost, and have expanded discussion in the broader impacts section of the potential drawbacks of using complex,
uninterpretable models for explanation.

• **(R3) (R4)** The question of how mismatched explainer model capacity affects results is indeed both interesting and
important. We address this in two ways. First, we are currently performing an experiment in which latent space
dimension and VAE architecture complexity are swept for a fixed classifier complexity. We will add these results
(presented both qualitatively with sample explanations, and quantitatively as in Figure 5(a-b) and Supplement Figure
11) along with a brief discussion in the main text. Second, we can use results from Feder & Merhav (*IEEE Trans.
Info. Theory*, 1994) to bound the capacity mismatch of our explainer (i.e., explainer error in predicting classifier
outputs) with the $I(\alpha; \widehat{Y})$ part of our objective. In practice, this result means that a sufficiently large value of $I(\alpha; \widehat{Y})$
serves as a certificate that the explainer complexity is sufficient to explain the classifier. We have added an appendix
containing details and implications of this analysis and a brief discussion to the conclusion of the main paper.

[Meta-Review · NeurIPS 2020]

This paper presents a generative model to "explain" any given black-box classifier and its training dataset. Explanation is through a hidden factor that can control or intervene in the output of the classifier. The discovery is based on a objective with two terms: 1) a proposed Information Flow that denotes the causal effect from the hidden factor to the classifier output and 2) a distribution similarity to impose the discovered hidden factor can generate back the feature space. Reviewers found this a borderline paper. After the discussion phase all reviewers are leaning towards acceptance. They pointed out as strengths that this is a very well-written paper, presenting a simple yet effective method, with extensive ablative experiments. However, they also pointed out several weaknesses, including a somewhat overstated storyline, which frames the method into causal inference when this does not seem to be really needed (as R1 points out and R2 agrees, their setup “renders almost all techniques of causal inference such as do-operation and counterfactuals trivial”). The author response alleviated some concerns (in particular from R1 and R4), but the feeling that causality is not really needed here still remains. The authors need to make it clearer why mutual information is a good metric for causal influence (as pointed out by R1 and R2), since MI measures a type of correlation. R3 also points out that “the theoretical results, propositions 2 and maybe also 3, are almost immediate observations rather than "surprising" results.“ It's not clear how the method would scale to problems where the classifier is complex and derives its output from many causally relevant factors. A discussion about this should be added (see R3’s comments). Overall, this paper could be accepted but it is not outstanding and it needs considerable revision (doable in camera ready time) to incorporate the necessary improvements. I lean towards accepting but urge the authors to follow the reviewers’ suggestions to improve the paper. In particular, I second R1’s suggestion to update the storyline and compare and discuss the related work in disentanglement, which seems to be more linked to the main contribution of this paper.